# Large-scale group-hierarchical DEMATEL method for complex systems

**Wenyu Chen**[1,2]*, **Weimin Li**[2]◉, **Lei Shao**[2]◉, **Tao Zhang**[3]◉, **Xi Wang**[4]◉

**1** Graduate Collage, Air Force Engineering University, Xi'an, China, **2** Air and Missile Defense College, Air Force Engineering University, Xi'an, China, **3** Unit 94907 of PLA, Nanchang, Jiangxi, China, **4** Xi'an Satellite Control Center, Xi'an, China

◉ These authors contributed equally to this work.
* lorilouto_cwy@163.com

**Data Availability Statement:** Detailed data information in Appendix A-D is provided on https://osf.io/gxtj5.

**Funding:** This research was funded by National Natural Science Foundation of China, grant number

## Abstract

Existing Decision-Making Trial and Evaluation Laboratory (DEMATEL) methods are mostly suitable for simple systems with fewer factors, and lack effective integration of expert knowledge and experience from large-scale group populations, resulting in a potential compromise of the quality of the initial direct relation (IDR) matrix. To make DEMATEL better suited for the identification of critical factors in complex systems, this paper proposes a hierarchical DEMATEL method for large-scale group decision-making. Considering the limitations of expert knowledge and experience, a method based on expert consistency network for constructing the expert weight matrix is designed. The expert consistency network is constructed for different elements, and the weights of experts in different elements are determined using the clustering coefficient. Following the principles of the classic DEMATEL method, the steps for identifying key elements in complex systems using the large-scale group-hierarchical DEMATEL method are summarized. To objectively test the effectiveness and superiority of the decision algorithm, the robustness of the algorithm is analyzed in an interference environment. Finally, the superiority of the proposed method and algorithm is verified through a case study, which demonstrating that the proposed decision-making method is suitable for group decision-making in complex systems, with high algorithm stability and low algorithm deviation.

## Introduction

In the knowledge economy era, the number of influence factors for complex organizations and systems is increasing, and the complex interrelationships between factors lead to complexity and uncertainty in management of complex organizations and systems. Accurately identifying key factors and clarifying the importance between factors have become important research topics in organizational management and decision-making, leading to the emergence of a series of evaluation and decision-making methods [1–3].

Many Multi-Criteria Decision Making (MCDM) methods are applied to identify influence factors by determining the weights of the criteria, including commonly used ones such as: CRiteria Importance Through Intercriteria Correlation (CRITIC) [4–6], BWM(Best-Worst Method) [7], Complex Proportional Assessment (COPRAS) [8], Simultaneous Evaluation of

62173339, 61873278. The funders had no role in study design, data analysis, decision to publish, or preparation of the manuscript.

Criteria and Alternatives(SECA) [9], Combinative Distance-based Assessment (CODAS) [10], Stepwise Weight Assessment Ratio Analysis (SWARA) [11], Method based on the Removal Effects of Criteria (MEREC) [12] and Evaluation based on Distance from Average Solution (EDAS) [13].

Professors Gabus and Fontela proposed the Decision-Making Trial and Evaluation Laboratory (DEMATEL) algorithm [14] in the 1970s. The algorithm identifies key elements by constructing a complex interrelationship diagram between factors and is used for analyzing complex systems.

Compared to the other methods, the DEMATEL method, based on graph theory, provides a more intuitive representation of the complex interrelationships between factors and its calculation process is more simple and straightforward. It has been widely promoted and applied in many fields such as system engineering, and management science. Costa Federica et al. investigated the role of human factors in promoting the establishment of sustainable continuous improvement (SCI) environment [15] by the DEMATEL, Huang et al. analyzed the key elements of circular supply chain management (CSCM) [16], Shahriar et al. analyzed the complex influence factors in the Covid-19 vaccine supply chain [17]. In addition, there are many application cases, which are not listed one by one here. Furthermore, many scholars have improved the traditional DEMATEL method. Some researchers are committed to combining the DEMATEL with other methods. Mohammad et al. improved DEMATEL by combining it with the best-worst method (BWM) and Bayesian network (BN), and applied it to safety management [18]. Sirous et al. studied the problem of selecting technical suppliers by constructing the Delphi-DEMATEL-ELECTRE method [19]. In addition to considering the combination with other methods, some researchers have improved the DEMATEL scale method and expanded the expression of expert judgment information, mainly about point estimation judgment information [20, 21], fuzzy number estimation information [22–25], and grey number estimation information [26–28]. In addition, some researchers have studied the normalization processing of the decision matrix in DEMATEL to solve the problem that the normalized matrix does not converge in some cases. Michnik et al. proposed a new DIM normalization processing method to solve the problem of the unsolvable TIM matrix [29], and Chen et al. expanded the normalization method for the influence matrix in DEMATEL [30].

However, both DEMATEL method and its improved version require experts to analyze each factor in pairs. In a complex system with $n$ factors, experts need to make $n(n-1)$ judgments. When $n$ is larger than 10, this will require a significant workload, leading to emotional fatigue and boredom that may affect the decision-making results. This limitation restricted the application of DEMATEL in complex systems with numerous factors, and it was not given enough attention by researchers. In 2021, Du proposed a hierarchical DEMATEL method [31], which adopts hierarchical decomposition to divide complex systems into several subsystems. Experts only need to make judgments on subsystems, and then use the proposed method for data integration. This method greatly simplifies the DEMATEL calculation steps and processes in complex systems, reduces the workload of experts, and has attracted the attention of many scholars. Moreover, considering the limitations of individual expert knowledge and experience, Du further introduced group decision-making based on the hierarchical DEMATEL [32], however, the group decision-making method is only suitable for small expert groups. When dealing with complex system identification, a large-scale expert group with not less than 20 people is usually needed [33, 34]. In addition, when a large-scale expert group makes decisions, the rationality of expert weights directly influences the accuracy of decision-making results. However, the weight of experts in Du's study is subjective, lacking persuasive expert weight calculation. The references mentioned above were subjected to comparative analysis, as shown in Table 1.

**Table 1. Application of the DEMATEL method.**

| Papers | Method | Application Scenario | Innovative approach |
|---|---|---|---|
| [15, 21] | Classic DEMATEL | Investigate the role of human factors in promoting the establishment of sustainable continuous improvement (SCI) environment; Identify the key factors affecting the supply chain in the electronics industry | Application of classical method |
| [16, 20] | AHP+DEMATEL | Assessing critical success factors for circular supply chain management (CSCM) implementation of blockchain; Explore the key factors influencing stock price behavior | Methods composition application |
| [17] | IFS+DEMATEL | Analyses have been conducted on the critical challenges of the COVID-19 vaccine supply chain | Methods composition application |
| [18] | BWM+BN +DEMATEL | Identifying the impact of risk factors and sources of information on the decision-making process | Methods composition application |
| [19, 28] | Gray DEMATEL | Studying the causal relationships of influencing factors in the decision-making process | Methods composition application |
| [22, 25] | Fuzzy DEMATEL | Estimate and map the suitability classes of ecotourism potentials in the study area of "Dunayski kljuc" region (Serbia); Analyzing the facilitating factors for supply chain responsiveness | Methods composition application |
| [26] | Gray DEMATEL +ANP | Explores favorable methods to evaluate the green mining performance (GMP) of underground gold mines | Methods composition application |
| [30] | DEMATEL | A new matrix normalization method has been researched and proposed | Innovation in Method |
| [31, 32] | Hierarchical DEMATEL | The hierarchical DEMATEL method has been proposed to make the DEMATEL method applicable to complex systems with many factors; based on the proposed hierarchical DEMATEL method, a program for small-group experts to reach consensus has been designed | Innovation in Method |

In summary, although hierarchical DEMATEL has improved the shortcomings of traditional DEMATEL method and can effectively analyze factors in complex systems, how to integrate and utilize the wisdom of large-scale expert groups and improve the scientificity of the IDR matrix of hierarchical DEMATEL method is a new and urgent problem to be solved. This article believes that as the number of factors in complex systems increases and the relationships between factors become more complex, decision making by a small group of experts may not be sufficient to cope with such complexity. It is necessary to determine the direct influence matrix in the hierarchical DEMATEL method by a large group of experts. Decision making by a large group of experts presents the following characteristics:

1. The group size is relatively large, usually consisting of no fewer than 20 decision-making experts;

2. The decision-making problem exhibits multidimensional, complex, and stochastic attributes;

3. High consistency requirements need to be met among the group.

When solving problems of large-scale group decision-making, the main difficulties are as follows [35]:

1. There are significant differences among decision-makers. It is necessary to identify the status of each decision-maker and assign corresponding weights to achieve scientific evaluation results.

2. Due to the large size of the decision-makers, it is important to use effective methods to gather the opinions of large groups to avoid leverage effects caused by intentional praise or criticism during the evaluation process.

3. When group opinions are relatively scattered, it is necessary to effectively coordinate the differences in preferences among decision-makers to maximize the satisfaction of large-scale group decisions.

In the problem of large-scale group decision-making, how to objectively determine the weight of each expert is a key issue. However, this issue is often overlooked by researchers. Chen pointed out that only 41% of the cited papers in group decision-making problems mentioned the determination of expert weights [36].

There are three main methods for determining expert weights in existing research: subjective method, objective method, and comprehensive method. For the subjective method, expert weights are calculated based on factors such as age, attitude, and experience, and the mutual evaluation of experts [37]. Multiplicative Analytic Hierarchy Process (MAHP) [38], Simple Multi-Attribute Rating Technique (SMART) [39], and Delphi [40] are key methods for subjective expert weight determination. The objective method is based on the evaluation performance of experts, using individual decision matrices (IDMs) proposed by each expert as the main basis for judgment, and assigning different weights, which is usually more objective [41]. The weight is mainly determined based on the degree of closeness between the expert's individual decision and the group decision [42–44]. In addition, some researchers have constructed expert opinion adjustment mechanisms, attempting to achieve consistency of group opinions by adjusting experts' weights or decisions as much as possible. Pang developed a nonlinear programming model and determined expert weights by maximizing group consensus [45], where expert weights were adaptively adjusted based on their decisions. Yang used a fixed-point iteration method to adjust expert weights multiple times [46].

However, from the relevant research on expert weights, it can be seen that when the research problem is a multi-attribute decision-making problem, almost all methods use a weight value to represent the expert's evaluation status for all attributes. For example, literatures [46–48] all use a weight to represent the expert's performance in all fields.

In reality, each expert has limited knowledge and abilities in their own expertise. Using a single weight to represent an expert's status in all fields is unreasonable, which cannot fully reflect the expert's professionalism to highlight their important position in their research field. Repeatedly iterating to solve expert weights to achieve consensus of group opinions often puts pressure on experts who adhere to their own opinions, forcing them to give up their decisions and ideas.

Finally, since decision-making problems are subjective progresses, the results of decisions do not have a correct answer, only subjective judgments of "reasonable" or "unreasonable". It's hard to judge whether one's method is superior to others. Most studies usually compare the decision results of their proposed method with other methods through numerical calculation examples. When inconsistencies occur, researchers often use subjective analysis to explain the rationality of their method, which often lacks persuasiveness. How to use a verification experiment to demonstrate the superiority of the proposed method instead of subjective analysis and judgment is an aspect that almost no researchers have studied.

Therefore, based on the above analysis, we find that the existing research on using DEMATEL method to identify factors in complex systems has the following shortcomings:

1. The traditional DEMATEL method can only be applied to situations with fewer elements. When the number of elements n>10, it will significantly increase the number of expert judgments and workload.

2. Although Du attempted to solve problem (1) by using group hierarchical DEMATEL method, the expert group was small and could not better reflect the wisdom of the expert group. In addition, the expert weight was subjectively given and lacked convincing objective calculation.

3. When calculating expert weights using the objective method, the importance of experts in all attribute fields is often measured by a single weight value. In reality, experts often have a

certain disciplinary background, and the degree of specialization of their judgment may be higher for some factors in complex systems, but lower for other factors. Using a single weight to determine the expert's judgment status for all factors is not scientific, and a more targeted method should be adopted.

4. There is a lack of convincing means to test the effectiveness and superiority of decision-making methods. Almost all studies subjectively analyze the differences in decision results between different methods when demonstrating the superiority of their methods. This inevitably leads authors to analyze in a way that favors their own methods and lacks persuasiveness.

Based on this, the author proposes a hierarchical DEMATEL method for large-scale group decision-making to identify key factors in complex systems and address the following issues:

1. In response to the heavy workload for experts in identifying complex systems using traditional DEMATEL methods, the hierarchical DEMATEL method proposed by Du is used to reduce the workload of experts in identifying elements of complex systems. The hierarchical DEMATEL method is combined with the large-scale group decision-making, and the expert number is not less than 20, which improves the quality of decision-making.

2. In response to the problem of expert weight solving, considering the potential influences of experts' knowledge, background, and profession, the factor setting weights are distinguished to construct the weight matrix of expert decision-making. Based on the IDR matrix of experts, the expert consistency network for a certain factor is constructed based on the performance of experts in scoring this factor, combined with the clustering coefficient of the weighted network to represent the consistency of experts. The weight of experts in scoring this factor is determined by their contribution to the clustering coefficient of the weighted network, and the weight matrix of expert decision-making is formed for all factors. This method avoids measuring the performance of experts in scoring in all fields with a single weight value, and can well represent the consistency of the group.

3. In order to objectively demonstrate the effectiveness of the proposed method, interference scenarios are set to analyze the robustness of the decision-making algorithm, and subjective analysis of the decision-making results is avoided. The stability and deviation of the decision-making method after interference are analyzed. The interference scenario refers to the implementation of interference on the original expert data of a certain scale to simulate the judgment deviation of experts. The stability index refers to the degree of change in the decision-making result after random interference, and the deviation index refers to the degree of deviation of the decision-making result after interference from the true value. Obviously, when the stability of the algorithm is high, it indicates that the decision-making algorithm will not easily change the decision-making result due to the influence of disturbance. When the deviation of the algorithm is low, it indicates that the decision-making algorithm can ensure a result closer to the true decision even if it is interfered with.

The innovations of this article are:

1. Introducing a new method for identifying the weights of experts in large-scale groups. This method assigns different weights to different indicators, abandoning the practice of using a single weight value to represent the decision-making status of experts under all indicators, in order to address the unique characteristics of each expert in terms of knowledge, skills, experience, and personality.

2. Using the network clustering coefficient to describe the consistency of expert groups in scoring the same indicator, and calculating expert weights through the consistency between

experts and the group, to maximize the requirement of opinion consistency in large-scale group decision-making.

3. The methods involved are more suitable for analyzing the correlation between various factors within complex systems and identifying key factors. It can not only reduce the workload of experts but also improve the scientificity of decision-making results.

4. Instead of analyzing the effectiveness of decision-making algorithms through subjective methods as in other studies, this article constructs interference scenarios to analyze the stability and bias of algorithm results when expert decision-making data is interfered with, which is more convincing.

The rest of this study is organized as follows. Section Preliminaries introduces the basic knowledge, mainly including the introduction of traditional DEMATEL method and hierarchical DEMATEL method. Section The Proposed method mainly including the construction of expert consistency network, calculation of weighted network clustering coefficient, calculation of expert weight matrix, collection of expert opinions, and overall calculation steps. In Section Case presentation and Methodology analysis, stability and deviation indicators of the algorithm are introduced, and numerical calculations and comparative analysis of different methods are performed. In Section Conclusion, our conclusions, contributions and innovations are explained.

## Preliminaries

This section introduces the DEMATLE method and the hierarchical DEMATEL method. The hierarchical DEMATLEL method is a new method based on the DEMATEL method proposed in [31], which decomposes the complex system into several subsystems, invites experts to score the degree of influence between elements within each subsystem, and finally turn the set of IDR matrices of all subsystems into a super IDR matrix. This method can effectively reduce the workload of experts and consider the hierarchical characteristics of complex systems. Each of these two methods is described below.

### DEMATEL method

The DEMATEL method is a structural model expansion method used to establish and analyze the interactions between complex criteria and oriented to factor analysis of complex systems, the basic elements of the DEMATEL method are as follows.

1. Determine the IDR matrix between the elements
   Suppose a system $F$ contains $N$ elements, denoted as $F = \{f_1, f_2, \cdots f_N\}$, and experts are invited to judge the degree of direct influence among these $N$ elements using a scale of $\{0,1,2,3,4\}$, representing "no influence", "low influence", "medium influence", "strong influence", and "very strong influence", respectively. The degree of influence of element $f_i$ on element $f_j$ is recorded as $x_{ij} \in \{0, 1, 2, 3, 4\}, i, j = 1, 2, \ldots, N$, the IDR matrix $\mathbf{X} = [x_{ij}]_{N \times N}$ is constructed according to $x_{ij}$, and when $i = j$, $x_{ij} = 0$, it represents no influence of the element itself.

2. IDR matrix normalization
   Normalize the IDR matrix constructed by experts [49]

$$\theta = \max(\max_{1 \leq i \leq N} \sum_j x_{ij}, \varepsilon + \max_{1 \leq j \leq N} \sum_i x_{ij}) \tag{1}$$

$$\mathbf{H} = [h_{ij}]_{N \times N} = \mathbf{X}/\theta \tag{2}$$

$\varepsilon$ is a non-Archimedean infinitesimal, the role of $\varepsilon$ is to ensure that the infinite powers of the normalized IDR matrix can converge to zero in order to satisfy the conditions for the subsequent third step of constructing the comprehensive influence matrix.

3. Constructing the comprehensive influence matrix
   The comprehensive influence matrix $\mathbf{T}$ is

$$\mathbf{T} = (t_{kl})_{n \times m} = \lim_{r \to \infty}(\mathbf{H}^1 + \mathbf{H}^2 + \cdots \mathbf{H}^r) = \mathbf{H}(\mathbf{I} - \mathbf{H})^{-1} \tag{3}$$

4. Calculation of causality and centrality of each factor
   Calculate $r_i = \Sigma_j t_{ij}$ and $d_j = \Sigma_i t_{ij}$ based on the comprehensive influence matrix $\mathbf{T}$. The sum of each row element of matrix $\mathbf{T}$, denoted by $r_i$, represents the sum of the influence of factor $f_i$ on other factors, which is called the influence degree of factor $f_i$. The sum of each column, denoted by $d_j$, represents the sum of the influence of other factors on factor $f_i$, which is called the being influenced degree of factor $f_i$.
   Let $r_i + d_i$ be the centrality of $f_i$, which characterizes the relative importance of factor $f_i$ in the system. Let $r_i - d_i$ be the causality of factor $f_i$. If $r_i - d_i > 0$, then $f_i$ is a causal factor; if $r_i - d_i < 0$, then $f_i$ is a receive factor.

5. Calculate the weights of each factor
   The weight of factor $f_i$ is

$$w_i = \frac{\sqrt{(r_i + d_i)^2 + (r_i - d_i)^2}}{\sum_{i=1}^{N} \sqrt{(r_i + d_i)^2 + (r_i - d_i)^2}} \tag{4}$$

$w_i$ satisfies $w_i \in [0,1]$, $\sum_{i=1}^{N} w_i = 1$.

In summary, the traditional DEMATEL method can be summarized in the following steps.

Step 1, experts are invited to make decisions on the system elements and construct the IDR matrix $\mathbf{X}$;.

Step 2, the IDR matrix is normalized, and the normalization matrix $\mathbf{H}$ is obtained by Eq (2);

Step 3, the comprehensive influence matrix $\mathbf{T}$ is constructed by Eq (3);

Step 4, the centrality and causality of the elements are calculated from the comprehensive influence matrix $\mathbf{T}$.

Step 5, the relative importance of each factor is calculated by Eq (4).

## Hierarchical DEMATEL method

The traditional DEMATEL method requires the expert to compare each element pairwise, which is suitable for simple systems with few elements. However, when there are many elements in the system, determining the IDR matrix requires a huge amount of work (if there are $n$ elements in the system, experts need to make $n(n-1)$ judgments). This can easily lead to expert mental fatigue and boredom. In addition, complex systems generally have hierarchical characteristics, which cannot be reflected in the traditional DEMATEL method. To address these issues, Du proposed the hierarchical DEMATEL method in reference [31], which is suitable for identifying key elements in complex systems that contain many system factors and have hierarchical characteristics among them. The method mainly includes the following contents:

1. Hierarchical decomposition of the system

   Hierarchical decomposition mainly includes vertical decomposition and horizontal decomposition. Horizontal decomposition focuses on dividing the critical factor identification problem of complex systems into several simple problems and vertical decomposition focuses on dividing the complex system into multi-level subsystems under a specific rule. Horizontal decomposition provides the rules for making vertical decomposition [31]. The complex system $F$ is decomposed into subsystems according to the horizontal and vertical decomposition. As shown in Fig 1, the complex system $F$ can be decomposed into subsystems $F_1 \sim F_Q$, and $F_1 \sim F_Q$ can be decomposed into subsystems $f_1^1 \sim f_{N_1}^1, f_1^q \sim f_{N_q}^q$, etc. If $F = \{f_1, f_2, \cdots f_N\}$, the decomposition stops when the system $F$ is decomposed downward to the factors $f_i \in \{f_1, f_2, \cdots f_N\}$, which is a component of $F$.

2. The IDR matrix of subsystem

   The experts are arranged to score all the subsystems, such as $F$, $F_1 \sim F_Q$ and $f_1^1 \sim f_{N_1}^1$, to obtain the IDR matrix of each system. For example, when system $F$ contains $Q$ subsystems, the IDR matrix of system $F$ is denoted as $\mathbf{X} = [x_{qq'}]_{Q \times Q}$, $x_{qq'}$ refers to the degree of direct influence of the $q$ subsystem in $F$ on the $q'$ subsystem; similarly, the IDR matrix of system $F_q$ containing $N_q$ subsystems can be denoted as $\mathbf{X}_q = [x_{nn'}^q]_{N_q \times N_q}$, and the superscript $q$ of $x_{nn'}^q$ is used to denote the system to which it belongs.

3. Calculate the super IDR matrix of the total system

   The super IDR matrix of the total system $F$ is obtained by integrating all the subsystem direct influence matrices after correction, which indicates the degree of direct influence among all elements. The integration rules are

$$\bar{\mathbf{X}} = [\bar{x}_{ij}]_{N \times N} = \begin{bmatrix} \bar{\mathbf{X}}_{11} & \cdots & \bar{\mathbf{X}}_{1Q} \\ \vdots & \ddots & \vdots \\ \bar{\mathbf{X}}_{Q1} & \cdots & \bar{\mathbf{X}}_{QQ} \end{bmatrix}$$

$$= \begin{bmatrix} \begin{bmatrix} \bar{x}_{11}^{11} & \cdots & \bar{x}_{1N_1}^{11} \\ \vdots & \ddots & \vdots \\ \bar{x}_{N_1 1}^{11} & \cdots & \bar{x}_{N_1 N_1}^{11} \end{bmatrix} & \cdots & \begin{bmatrix} \bar{x}_{11}^{1Q} & \cdots & \bar{x}_{1N_Q}^{1Q} \\ \vdots & \ddots & \vdots \\ \bar{x}_{N_1 1}^{1Q} & \cdots & \bar{x}_{N_1 N_Q}^{1Q} \end{bmatrix} \\ \vdots & \ddots & \vdots \\ \begin{bmatrix} \bar{x}_{11}^{Q1} & \cdots & \bar{x}_{1N_1}^{Q1} \\ \vdots & \ddots & \vdots \\ \bar{x}_{N_Q 1}^{Q1} & \cdots & \bar{x}_{N_Q N_1}^{Q1} \end{bmatrix} & \cdots & \begin{bmatrix} \bar{x}_{11}^{QQ} & \cdots & \bar{x}_{1N_Q}^{QQ} \\ \vdots & \ddots & \vdots \\ \bar{x}_{N_Q 1}^{QQ} & \cdots & \bar{x}_{N_Q N_Q}^{QQ} \end{bmatrix} \end{bmatrix} \tag{5}$$

where the calculation rules for the elements are

$$\bar{x}_{ij}^{qq'} = \begin{cases} \dfrac{x_{qq'}}{\sum_i \sum_j x_{ij}^{qq'}} x_{ij}^{qq'}, q = q' \\ \dfrac{z_i^q z_i^{q'}}{\sum_i \sum_j z_i^q z_i^{q'}} x_{qq'}, q \neq q' \end{cases} \quad for\ i = 1, \ldots, N_q, j = 1, \ldots, N_{q'} \tag{6}$$

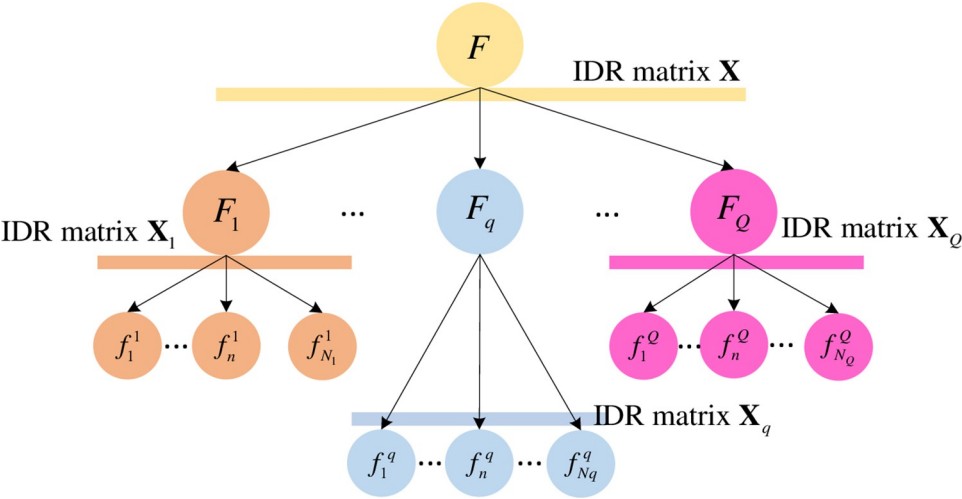

**Fig 1. Schematic diagram of subsystem division, this figure is from Du's literature.**

When $q = q'$, which means that the two subsystems are identical, $x_{ij}^{qq'} = x_{ij}^q$, the elements can be obtained directly from the expert scoring matrix for that system;

When $q \neq q'$, it means that the two subsystems are different, $x_{qq'}$ represents the degree of direct influence of the subsystems $F_q$ and $F_{q'}$, the subscript number represents the order of the subsystems $F_q$ and $F_{q'}$ in the IDR matrix of their superior system.

$z_i^q$ represents the centrality of factor $f_i^q$ in subsystem $F_q$, i.e.,

$$z_i^q = r_i^q + d_i^q = \sum_{i'} t_{ii'}^q + \sum_{i'} t_{i'i}^q \tag{7}$$

$t_{ii'}^q$ and $t_{ii'}^q$ are the elements in the comprehensive influence matrix $\mathbf{T}^q$ of the subsystem $F_q$ with $\mathbf{T}^q = [t_{ii'}^q]_{N_q \times N_q}$ and $\mathbf{T}^q$ is normalized according to the IDR matrix $\mathbf{X}_q$ of the subsystem $F_q$ using steps (2) to (3).

For the convenience of example, the above describes the simple case of two levels. When the level decomposition of a complex system involves multiple levels, the modified IDR matrix of the subsystem needs to be derived sequentially from the bottom to the top level, and the recursive integration from low to high forms the super IDR matrix, and the specific process is detailed in the literature [31].

4. Calculation of elemental importance and centrality

   $\bar{\mathbf{X}}$ is brought into the DEMATEL method as the IDR matrix, as in steps (1) to (5) in Section DEMATEL method, and the importance and centrality of each element are calculated.

## The proposed method

In this section, we introduce the hierarchical DEMATEL method to large-scale group expert decision making. We will construct an expert consistency network based on the performance of the large-scale group experts when scoring the same factors, express the consistency of the experts through the weighted network clustering coefficients, determine the expert weights using the contribution of each expert to the consistency, each expert will have different weights when scoring different factors, express the different abilities shown by the experts in different

fields through the weight matrix, and finally the resulting weighted IDR matrix of each system. The weighted IDR matrix will be integrated using the hierarchical DEMATEL method to obtain the centrality and the causality of each factor, and realize the large-scale group hierarchical DEMATEL decision.

On the one hand, this paper combines large-scale group experts with the hierarchical DEMATEL, proposes a new method of pooling group wisdom, and improves the authority and scientificity of the IDR matrix of subsystems; on the other hand, it integrates the ability of different experts in different fields, and highlights the professional status of experts in a certain field, because even some authoritative experts, who make judgments in certain fields, are not as scientific as experts who are good at that field.

## Problem description and hypothesis

The hierarchical DEMATEL method is used to make decisions on the importance of various elements within a complex system. Firstly, the complex system is divided into several subsystems according to levels and categories, and then $m$ experts are organized to judge the degree of influence among the subsystem factors to derive the IDR matrix. Due to the large number of levels and subsystems, the information of the parent system at each level is used to name a certain subsystem in the form of $F_{q_1 \supset \cdots \supset q_{p-1}}$, and $q_1 \ldots q_{p-1}$ represents the information of the parent system at each level.

For example, a subsystem with subscript $q_1 \supset q_2$, $q_2$ means it's serial number in its parent system, $q_1$ is the serial number for its parent system at the higher level, and so on. In Fig 2, system $F_{1 \supset 1}$ represents the subsystem of the second level, and its parent system is the first system of the first level. The IDR matrix given by the $n$th expert to the subsystem $F_{q_1 \supset \cdots \supset q_{p-1}}$ is denoted as $\mathbf{X}^{(n)}_{q_1 \supset \cdots \supset q_{p-1}}$, which includes pairwise comparisons of the degree of influence between the factors involved, using a scale of {0,1,2,3,4}, representing "no influence", "low influence", "medium influence", "strong influence", and "very strong influence". The interrelationships between the systems are shown in the Fig 2.

The problem to be solved in this paper is: How to fully explore the IDR matrix of $m$ experts on the subsystem analysis, determine the weight of each expert in scoring different factors, and obtain the weight matrix of experts to calculate the weighted IDR matrix $\bar{X}_{q_1 \supset \cdots \supset q_{p-1}}$ of the subsystem, and finally achieve the identification of the importance of each element for all factors based on the hierarchical DEMATEL method.

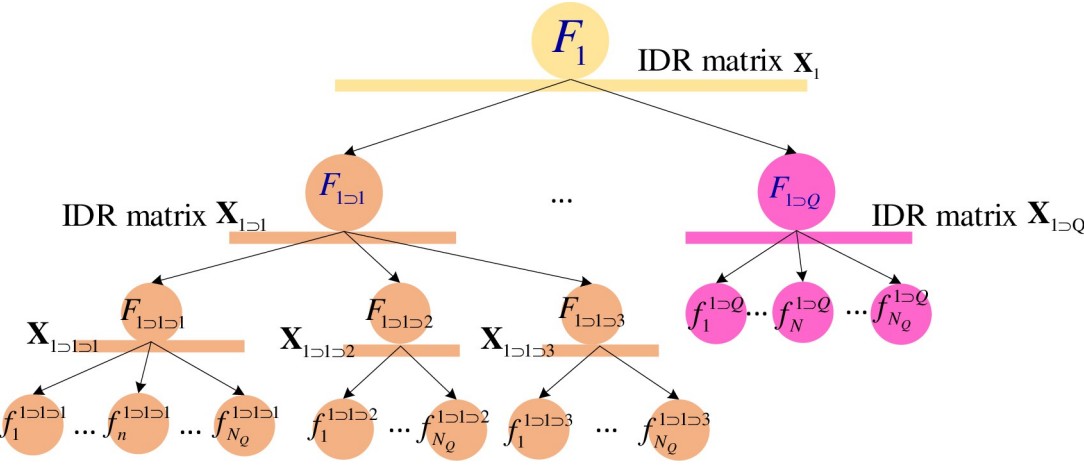

**Fig 2. Representation of multi-layer system with IDR matrix.**

### Building expert consistency networks

Suppose that $m$ experts are organized to score the subsystem $F_{q_1 \supset \cdots \supset q_{p-1}}$, which has $K$ elements, where the IDR matrix given by the $n$th expert is denoted as $\mathbf{X}^{(n)}_{q_1 \supset \cdots \supset q_{p-1}} \in \mathbf{R}^{K \times K}$,

$$\mathbf{X}^{(n)}_{q_1 \supset \cdots \supset q_{p-1}} = \begin{bmatrix} x^{(n)}_{11} & x^{(n)}_{12} & \cdots & x^{(n)}_{1K} \\ x^{(n)}_{21} & x^{(n)}_{22} & \cdots & x^{(n)}_{2K} \\ \vdots & \vdots & \ddots & \vdots \\ x^{(n)}_{K1} & x^{(n)}_{K2} & \cdots & x^{(n)}_{KK} \end{bmatrix} \tag{8}$$

The element $x^{(n)}_{ij}$ represents the influence degree of factor $i$ relative to factor $j$ in system $F_{q_1 \supset \cdots \supset q_{p-1}}$ made by the $n$th expert, the process of constructing the IDR matrix by the organization experts is shown in Fig 3.

Due to the differences in professional background and competence knowledge of experts, certain subjectivity and deviation will occur in scoring decisions, and different weights are needed to measure the importance of experts' decisions. In this paper, we use a weight matrix to determine the importance of experts for different factors instead of a single weight value, and establish an expert consistency network through the scoring performance of each expert to calculate the weight of experts for that factor.

**Definition 1:** Consistency of experts

Existing experts $a$ and $b$ make judgments on the degree of influence of factor $i$ on factor $j$ in system $F_{q_1 \supset \cdots \supset q_{p-1}}$. $x^{(a)}_{ij}$ is the decision value of expert $a$ and $x^{(b)}_{ij}$ is the decision value of expert $b$. Define the degree of agreement between experts $a$ and $b$ as

$$\sigma_{ab} = 1 - \frac{|x^{(a)}_{ij} - x^{(b)}_{ij}|}{T_N} \tag{9}$$

Where $a, b \in \{1, 2, \cdots m\}$, $T_N$ is the difference between the maximum and minimum values in the DEMATLE evaluation scale, and in scales $\{0, 1, 2, 3, 4\}$ of this paper, $T_N = 4$.

The rationality and nature of the above definition is discussed as follows:

**Property 1:**

$\sigma_{ab} \in [0, 1]$, when rating the importance of factor $i$ compared to factor $j$, if expert $a$ and $b$ give equal ratings, i.e., when $x^{(a)}_{ij} = x^{(b)}_{ij}$, $\sigma_{ab} = 1$, indicating that the two experts' agreement degree to reach the maximum value; when the difference between the ratings is the largest, i.e., when

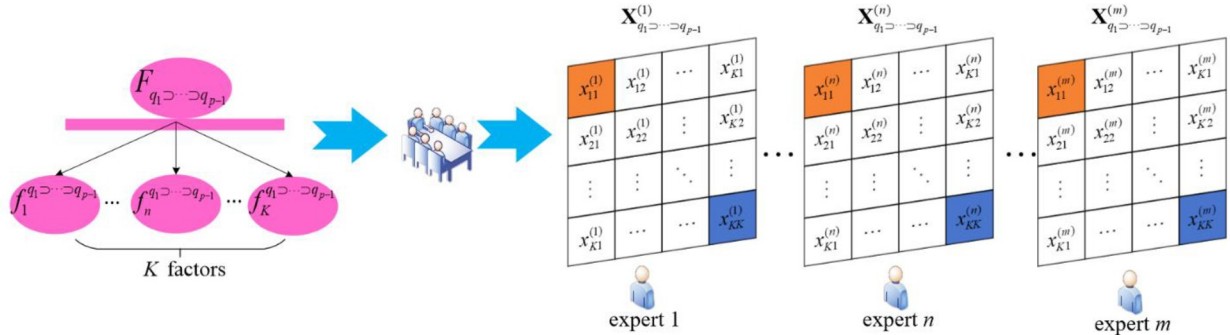

**Fig 3. Expert decision-making process.**

$|x_{ij}^{(a)} - x_{ij}^{(b)}| = T_N$, $\sigma_{ab} = 0$, indicating that the two experts' agreement degree reach the minimum value.

**Property 2:**

As the ratings given by the two experts become closer, the value of $\sigma_{ab}$ will be bigger, indicating that the agreement degree between two experts is also greater, and $\sigma_{ab} = \sigma_{ba}$, i.e., the relationship between the two decision makers is symmetric.

When scoring element $x_{ij}$ in the system $F_{q_1 \supset \cdots \supset q_{p-1}}$, an undirected weighted network is constructed based on the consistency exhibited by $m$ experts. The $m$ experts are the nodes in the complex network, and the agreement degree among experts is the weight of the edges. Assuming that $m$ experts score element $x_{ij}$ as $x_{ij}^{(1)}, x_{ij}^{(2)}, \cdots x_{ij}^{(m)}$, the consistency matrix for element $x_{ij}$ formed by $m$ experts is

$$\mathbf{Y}_{ij}^{q_1 \supset \cdots \supset q_{p-1}} = \begin{bmatrix} \sigma_{11} & \sigma_{12} & \cdots & \sigma_{1m} \\ \sigma_{21} & \sigma_{22} & \cdots & \sigma_{2m} \\ \vdots & \vdots & \ddots & \vdots \\ \sigma_{m1} & \sigma_{m2} & \cdots & \sigma_{mm} \end{bmatrix} \tag{10}$$

The matrix $\mathbf{Y}_{ij}^{q_1 \supset \cdots \supset q_{p-1}}$ can be considered as the consistency network adjacency matrix when $m$ experts score the element $x_{ij}$ in the system $F_{q_1 \supset \cdots \supset q_{p-1}}$. The network can be denoted as $G_{ij}^{q_1 \supset \cdots \supset q_{p-1}} = (V_{ij}^{q_1 \supset \cdots \supset q_{p-1}}, E_{ij}^{q_1 \supset \cdots \supset q_{p-1}})$. $G_{ij}^{q_1 \supset \cdots \supset q_{p-1}}$ is an undirected weighted network, $V_{ij}^{q_1 \supset \cdots \supset q_{p-1}} = \{v_1, v_2, \cdots v_m\}$ represents the set of $m$ experts as nodes, $E_{ij}^{q_1 \supset \cdots \supset q_{p-1}} = \{e_1, e_2, \cdots e_h\}$ represents the set of edges, and the weights of the edges are the agreement degree $\sigma_{ab}$ among the experts.

Similarly, the expert consistency network for all other elements can be constructed based on the expert scoring performance. Obviously, for subsystem $F_{q_1 \supset \cdots \supset q_{p-1}}$, since it contains $K$ elements, $K \times K$ consistency networks can be formed, and each network corresponds to an adjacency matrix, and the adjacency matrices of all networks will form the Super Consistency Matrix. The process of constructing an expert consistency network is shown in Fig 4.

The later section will effectively reveal the intrinsic connections among the expert members by further analyzing the consistency network of experts in order to clarify the weight of the experts' scores for each element.

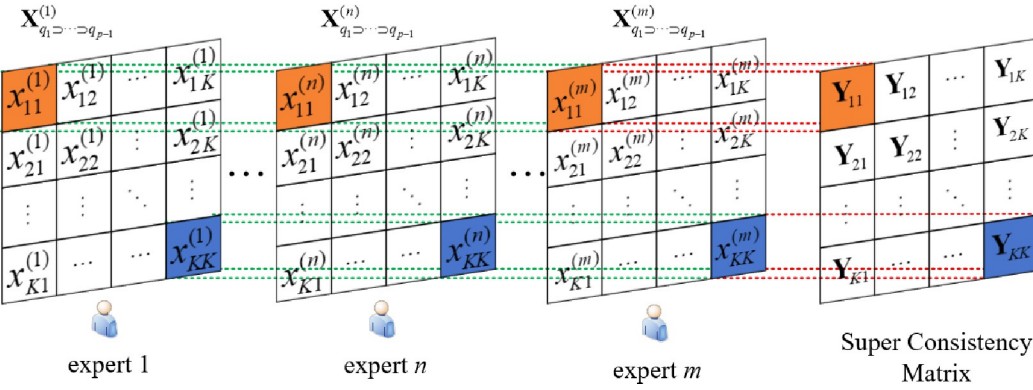

**Fig 4. Expert consistency network construction process.**

## Solving the weighted IDR matrix based on the weighted network clustering coefficient

After constructing experts consistency networks, we can study the relationship between experts based on the network features. The Clustering Coefficient of a network is the ratio of the connectivity between any two nodes in the network to the connectivity between the neighboring nodes they share.

In general, suppose a node $v_i$ has $k_i$ edges connecting it to other nodes, and these $k_i$ nodes are called neighbors of node $v_i$. Obviously, there are at most $C_{k_i}^2$ possible edges between these $k_i$ nodes. The ratio of the actual number of edges $E_i$ and the total number of possible edges $C_{k_i}^2$ between $k_i$ neighboring nodes of node $v_i$ is defined as the clustering coefficient $C_i$ of node $v_i$, i.e.

$$C_i = \frac{E_i}{C_{k_i}^2} \tag{11}$$

In simple terms, it is the ratio of the actual number of connections around a node to the theoretical maximum number of connections. The clustering coefficient $C$ of the whole network is the average of the clustering coefficients $C_i$ of all nodes $v_i$. That is

$$C = \frac{1}{N} \sum_{i=1}^{N} C_i \tag{12}$$

Clearly, when all nodes are isolated and there are no connecting edges, $C = 0$; and $C = 1$ only when the network is globally coupled, with any two nodes directly connected.

The clustering coefficient represents the degree of closeness and stability of groups formed in the network. When the clustering coefficient is higher, it indicates that the neighbors of the node are closer and the resulting clustering groups are more stable. Since this paper constructs a network based on the scores given by experts for a certain factor, it is only necessary to determine whether the clustering groups for the same factor judgment scenario are stable. The property of the clustering coefficient can just reflect the consistency of the expert group. The expert consistency network in this paper is a weighted network. Onnela [50] studied the clustering coefficient of weighted networks, and defined the clustering coefficient of nodes $v_i$ in a weighted network as:

$$C_i = \frac{2}{k_i(k_i - 1)} \sum_{j,k} (\tilde{w}_{ij} \tilde{w}_{jk} \tilde{w}_{ki})^{\frac{1}{3}} \tag{13}$$

In this equation, $w_{ij}$ represents the edge weight between node $v_i$ and $v_j$, $\tilde{w}_{ij} = w_{ij}/\max(w_{ij})$ is the normalized weight, and $k_i$ is the degree of node $v_i$. Combining Eqs (10) and (13), when all experts score the elements $x_{ij}$ in subsystem $F_{q_1 \supset \cdots \supset q_{p-1}}$, the clustering coefficient vector $C_{ij}^{q_1 \supset \cdots \supset q_{p-1}}$ of $m$ experts in weighted network $G_{ij}^{q_1 \supset \cdots \supset q_{p-1}}$ can be obtained based on the consistency matrix $\mathbf{Y}_{ij}^{q_1 \supset \cdots \supset q_{p-1}}$:

$$C_{ij}^{q_1 \supset \cdots \supset q_{p-1}} = [C_1, C_2, \cdots C_i, \cdots C_m] \tag{14}$$

Where

$$C_i = \frac{2}{k_i(k_i - 1)} \sum_{j,k} (\tilde{\sigma}_{ij} \tilde{\sigma}_{jk} \tilde{\sigma}_{ki})^{\frac{1}{3}} \tag{15}$$

$$\tilde{\sigma}_{ij} = \sigma_{ij}/\max(\sigma_{ij}) \tag{16}$$

We can perform exponential normalization on Eq (14) with the exponent $m$ being the number of experts, to obtain the weight value of the $n$th expert in network $G_{ij}^{q_1 \supset \cdots \supset q_{p-1}}$.

Normalize Formula (14) by exponentiation, where the exponent $m$ is the number of experts, to obtain the weight value $w_n$ of the $n$th expert in network $G_{ij}^{q_1 \supset \cdots \supset q_{p-1}}$, there is

$$w_n = \frac{(C_n)^m}{\sum\limits_{n=1}^{m} (C_n)^m} \tag{17}$$

To distinguish the subsystem information and elements targeted by the weight values, we add a superscript $q_1 \supset \cdots \supset q_{p-1}:ij$ to the weight values in Eq (17), so that the weight value of the $n$th expert when judging the degree of influence of factor $i$ on factor $j$ in system $F_{q_1 \supset \cdots \supset q_{p-1}}$ can be expressed as $w_n^{q_1 \supset \cdots \supset q_{p-1}:ij}$, where

$$w_n^{q_1 \supset \cdots \supset q_{p-1}:ij} = \frac{(C_n)^m}{\sum\limits_{n=1}^{m} (C_n)^m} \tag{18}$$

Repeating Eqs (11) to (18), we can obtain the weight matrix of experts for all factors in the system $F_{q_1 \supset \cdots \supset q_{p-1}}$. The weight matrix of the $n$th expert for the system $F_{q_1 \supset \cdots \supset q_{p-1}}$ is

$$\mathbf{W}_n^{q_1 \supset \cdots \supset q_{p-1}} = [w_n^{q_1 \supset \cdots \supset q_{p-1}:ij}]_{K \times K} \tag{19}$$

By combining the weight matrix and the IDR matrix, the weighted IDR matrix $\bar{\mathbf{X}}_{q_1 \supset \cdots \supset q_{p-1}}$ for subsystem $F_{q_1 \supset \cdots \supset q_{p-1}}$ is

$$\bar{\mathbf{X}}_{q_1 \supset \cdots \supset q_{p-1}} = \sum_{n=1}^{m} \mathbf{W}_n^{q_1 \supset \cdots \supset q_{p-1}} \odot \mathbf{X}_{q_1 \supset \cdots \supset q_{p-1}}^{(n)} \tag{20}$$

Based on Eqs (8) to (20), we can obtain the weighted IDR matrix for each subsystem by constructing an expert consensus network, calculating expert weights based on the weighted network clustering coefficient, and synthesizing the weighted IDR matrices for each subsystem. By using the weighted IDR matrices for each subsystem as the IDR matrix of the hierarchical DEMATEL method, we can analyze the causality and centrality of all elements in a complex system by the hierarchical DEMATEL method.

In summary, the main process of the proposed method in this article is shown in Fig 5.

## Calculation steps of the proposed method

Based on the previous description, the computational steps of the large-scale group decision hierarchical DEMATEL method proposed in this paper are sorted out as follows.

Step 1: To classify a complex system hierarchically

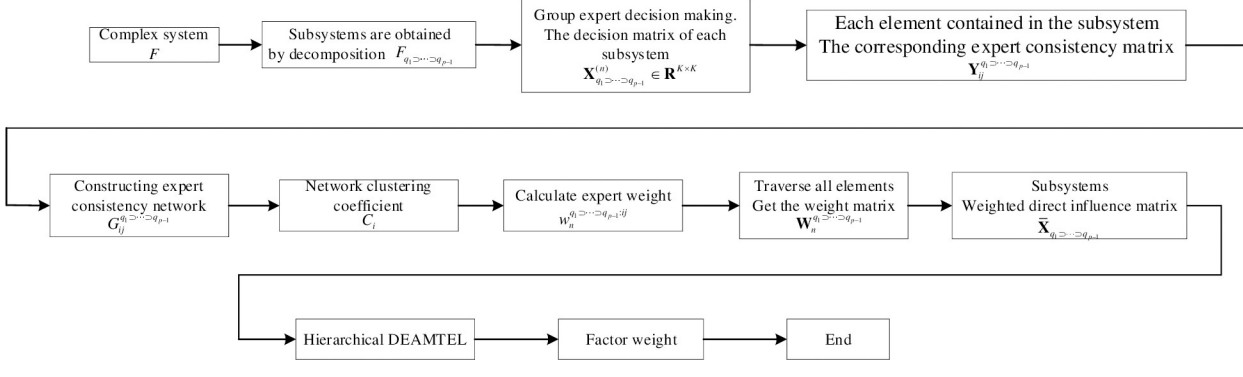

**Fig 5. Flow chart of the decision algorithm.**

For a system $F = \{f_i | i = 1, 2, \ldots, y\}$, the influence factors $f_i$ are divided into subsystems by hierarchy and by attributes, and the subsystems are denoted as $F_{q_1 \supset \cdots \supset q_{p-1}}$;

Step 2: Constructing the IDR matrix for each subsystem.

The IDR matrix obtained from the judgment of subsystem $F_{q_1 \supset \cdots \supset q_{p-1}}$ by the $n$th expert is $\mathbf{X}_{q_1 \supset \cdots \supset q_{p-1}}^{(n)}$;

Step 3: Constructing expert consistency networks for each factor

The expert consistency network $G_{ij}^{q_1 \supset \cdots \supset q_{p-1}} = (V_{ij}^{q_1 \supset \cdots \supset q_{p-1}}, E_{ij}^{q_1 \supset \cdots \supset q_{p-1}})$ is formed when the group experts judge the influence degree of factor $i$ on factor $j$ in the subsystem $F_{q_1 \supset \cdots \supset q_{p-1}}$. The network adjacency matrix is calculated as Eqs (9) to (10);

Step 4: Calculation of expert weight matrix by weighted network clustering coefficients

According to Eqs (13) to (16), the clustering coefficient vector $C_{ij}^{q_1 \supset \cdots \supset q_{p-1}}$ of all experts for the network $G_{ij}^{q_1 \supset \cdots \supset q_{p-1}} = (V_{ij}^{q_1 \supset \cdots \supset q_{p-1}}, E_{ij}^{q_1 \supset \cdots \supset q_{p-1}})$ is calculated, and the weights of all experts in this network are obtained by normalizing Eq (18); the consistency network for all factors is traversed, and the weight matrix $\mathbf{W}_n^{q_1 \supset \cdots \supset q_{p-1}}$ of the experts is solved;

Step 5: Calculate the weighted IDR matrix

Calculate the weighted IDR matrix $\bar{\mathbf{X}}_{q_1 \supset \cdots \supset q_{p-1}}$ for subsystem $F_{q_1 \supset \cdots \supset q_{p-1}}$ from Eq (20);

Step 6: Solving the super IDR matrix by hierarchical DEMATEL method

Using Eqs (5) to (6), the super IDR matrix is solved through weighted IDR matrix $\bar{\mathbf{X}}_{q_1 \supset \cdots \supset q_{p-1}}$;

Step 7: Using the traditional DEMATEL method to calculate the centrality and causality of each factor

According to Eqs (1) to (4), the centrality $r_i + d_i$ of factor $f_i$ is calculated, which characterizes the relative importance of factor $f_i$ in the system; the causality $r_i - d_i$ of factor $f_i$ is calculated, and if $r_i - d_i > 0$, then $f_i$ is the cause factor, and if $r_i - d_i < 0$, then $f_i$ is the receive factor.

## Case presentation and methodology analysis

### Case presentation

There are many influence factors for combat capability, and the relationships between these factors are complex and intertwined. The hierarchical characteristics of these factors are obvious, so combat capability is a typical complex system.

Identifying and analyzing the key factors that influence combat capability is crucial for improving it. Sixteen factors ($f_1 \sim f_{16}$) that influence combat capability can be identified and

**Table 2. Influence factors of combat capability.**

| Dimension | Factor | Content | Dimension | Factor | Content |
|---|---|---|---|---|---|
| Communication | $f_1$ | Interference and anti-interference ability | | $f_9$ | $C^2$ system response time |
| | $f_2$ | Signal transmission rate | Logistics | $f_{10}$ | Adequacy of war reserve |
| | $f_3$ | Signal transmission security | | $f_{11}$ | Equipment maintenance efficiency |
| Intelligence | $f_4$ | Intelligence collection efficiency | | $f_{12}$ | Materiel resupply capability |
| | $f_5$ | Accuracy of intelligence analysis | Fire support | $f_{13}$ | Speed of maneuver |
| Command | $f_6$ | Commander ability | | $f_{14}$ | Equipment protection capability |
| | $f_7$ | $C^2$ system intelligence degree | | $f_{15}$ | Killing accuracy |
| | $f_8$ | $C^2$ system compatibility | | $f_{16}$ | Attack speed |

classified into five dimensions: communication $F_{1\subset 1} = \{f_i | i = 1, 2, 3\}$, intelligence $F_{1\subset 2} = \{f_i | i = 4, 5\}$, command $F_{1\subset 3} = \{f_i | i = 6, 7, 8, 9\}$, logistics $F_{2\subset 1} = \{f_i | i = 10, 11, 12\}$, and fire support $F_{2\subset 2} = \{f_i | i = 13, 14, 15, 16\}$, as shown in Table 2. Based on their attributes, these five dimensions can be considered as belonging to the command and control and communication systems $F_1 = \{F_{1\supset q_2} | q_2 = 1, 2, 3\}$, as well as the fire and logistics support systems $F_2 = \{F_{2\supset q_2} | q_2 = 1, 2\}$. The detailed hierarchical relationships are shown in Fig 6.

Communication, intelligence, command, logistics, and fire support factors are all critical in combat, and there are complex interrelationships among the factors within each subsystem, as well as between different subsystems.

The proposed method used to identifying the key factors influencing combat capability. In this case, combat capability includes 16 factors, and using the traditional DEMATEL method would require experts to make 240 judgments, which is obviously a significant workload. However, using the hierarchical DEMATEL method only requires experts to make 55 judgments, which clearly demonstrates the advantages of this method.

Step 1 Hierarchical decomposition

The above elements are divided by hierarchy to form the structure diagram shown below

The combat capability system is decomposed into a two-level structure according to the hierarchy, with the first level containing two subsystems, command and control and communications $F_1$, and firepower and logistical support $F_2$. The second level contains 5 subsystems of communication $F_{1\supset 1}$, intelligence $F_{1\supset 2}$, command $F_{1\supset 3}$, logistics $F_{2\supset 1}$, and fire support $F_{2\supset 2}$. These five subsystems specifically contain these 16 specific factors.

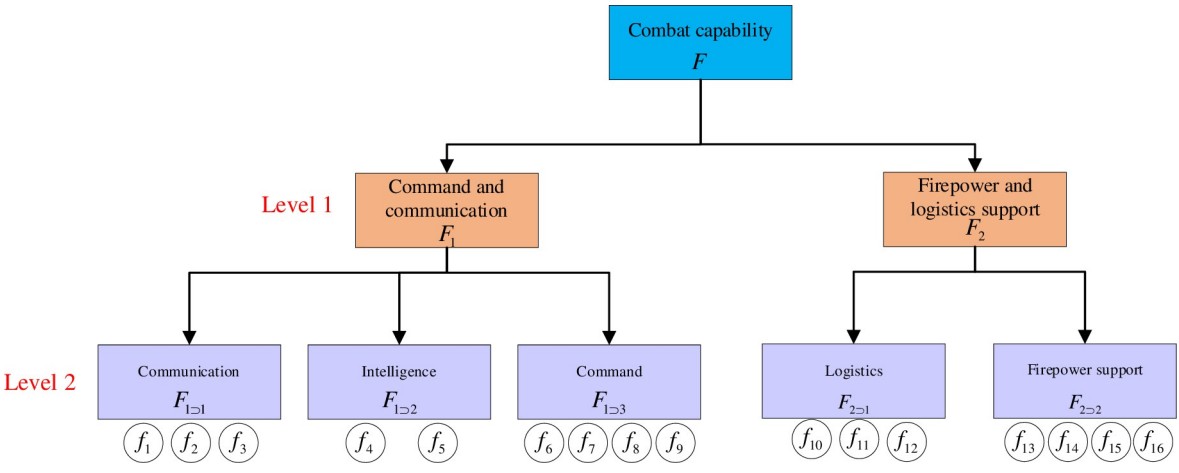

**Fig 6. Hierarchy of combat capability.**

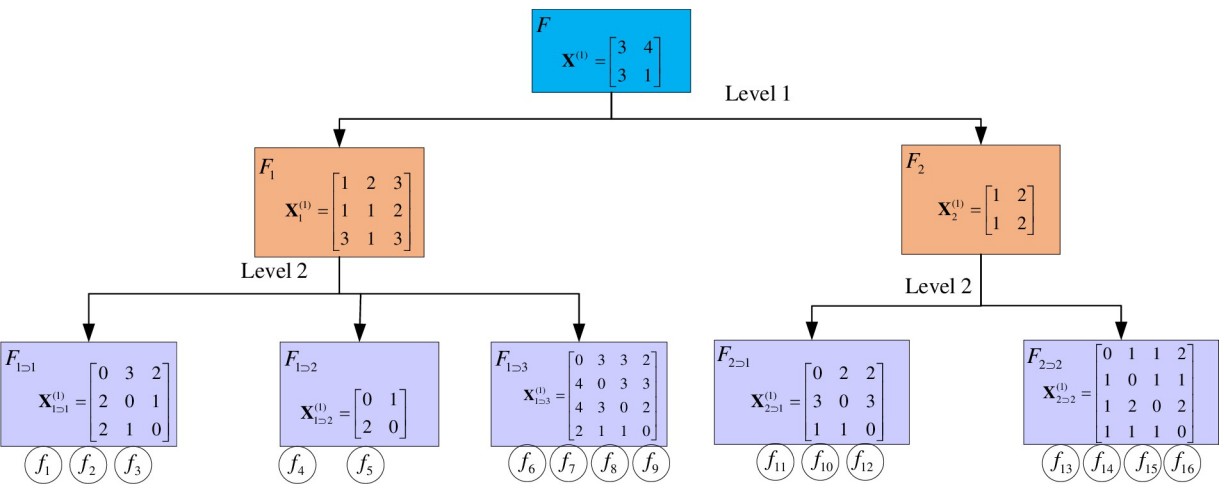

**Fig 7. The decision situation for expert 1.**

Step 2 Constructing the IDR matrix for each subsystem

The IDR matrix needs to invite experts to judge the influence relationship between the factors contained in the system, using the 0~4 scale method, 20 military theory researchers, weapon equipment professionals, combat commanders were invited to judge the system $F, F_1$, $F_2, F_{1 \supset 1}, F_{1 \supset 2}, F_{1 \supset 3}, F_{2 \supset 1}, F_{2 \supset 2}$, each expert needs to make 55 decisions, and the matrices $\mathbf{X}^{(n)}$, $\mathbf{X}_1^{(n)}, \mathbf{X}_2^{(n)}, \mathbf{X}_{1 \supset 1}^{(n)}, \mathbf{X}_{1 \supset 2}^{(n)}, \mathbf{X}_{1 \supset 3}^{(n)}, \mathbf{X}_{2 \supset 1}^{(n)}, \mathbf{X}_{2 \supset 2}^{(n)}$ respectively represent the IDR matrix obtained from the judgment of the $n$th expert on the corresponding system.

The decision situation of expert 1 is shown in Fig 7. Due to the large volume of data, please see **Appendix A** at https://osf.io/gxtj5 for additional expert decision information.

Step 3 Building expert consistency networks

Taking the construction of expert consistency network $G_{11}$ as an example, the naming rule of the expert consistency network indicates that the superscript of $G_{11}$ represents the system number, and the subscript represents the factor number. Therefore, the consistency network formed by all experts when judging the degree of influence of the first factor of system $F$ (i.e., system $F_1$) on itself.

The IDR matrix $\mathbf{X}^{(n)}$ of 20 experts is extracted, and the element $x_{11}^{(n)}$ $n = 1, 2 \ldots, 20$ in the first row and first column is calculated according to Formula (9) to obtain the consistency matrix $\mathbf{Y}_{11}$ formed by the 20 experts' scoring on this factor. The specific values corresponding to each row and column are shown in Table 3, and the expert consistency matrix for other elements can be found in **Appendix B** at https://osf.io/gxtj5.

In order to see the sparsity between experts more intuitively, according to the consistency matrix $\mathbf{Y}_{11}$ of the network $G_{11}$, we use Gephi to draw the structure of the consistency network $G_{11}$ as shown in Fig 8, even the thickness of the edge represents the weight of the edge, from the figure can be roughly seen that the experts are not equally close to each other.

Step 4 Calculation of expert weight matrix by weighted network clustering coefficients

Continue to use network $G_{11}$ as an example to illustrate. The clustering coefficients of the 20 experts for the network $G_{11}$ are calculated according to Eqs (13) to (16) in Table 4.

The weights of the experts are obtained by normalizing the clustering coefficients according to Eq (17), the normalization index $m = 20$, to serve the purpose of reducing the weights of the experts who are far from the group consensus and giving more weights to the experts with high consensus.

**Table 3. Consistency matrix $Y_{11}^1$.**

| Expert | 1 | 2 | 3 | 4 | 5 | 6 | 7 | 8 | 9 | 10 | 11 | 12 | 13 | 14 | 15 | 16 | 17 | 18 | 19 | 20 |
|---|---|---|---|---|---|---|---|---|---|---|---|---|---|---|---|---|---|---|---|---|
| 1 | 1 | 1 | 1 | 1 | 1 | 1 | 1 | 0.75 | 1 | 1 | 0.75 | 1 | 1 | 1 | 0.75 | 1 | 1 | 1 | 1 | 0.5 |
| 2 | 1 | 1 | 1 | 1 | 1 | 1 | 1 | 0.75 | 1 | 1 | 0.75 | 1 | 1 | 1 | 0.75 | 1 | 1 | 1 | 1 | 0.5 |
| 3 | 1 | 1 | 1 | 1 | 1 | 1 | 1 | 0.75 | 1 | 1 | 0.75 | 1 | 1 | 1 | 0.75 | 1 | 1 | 1 | 1 | 0.5 |
| 4 | 1 | 1 | 1 | 1 | 1 | 1 | 1 | 0.75 | 1 | 1 | 0.75 | 1 | 1 | 1 | 0.75 | 1 | 1 | 1 | 1 | 0.5 |
| 5 | 1 | 1 | 1 | 1 | 1 | 1 | 1 | 0.75 | 1 | 1 | 0.75 | 1 | 1 | 1 | 0.75 | 1 | 1 | 1 | 1 | 0.5 |
| 6 | 1 | 1 | 1 | 1 | 1 | 1 | 1 | 0.75 | 1 | 1 | 0.75 | 1 | 1 | 1 | 0.75 | 1 | 1 | 1 | 1 | 0.5 |
| 7 | 1 | 1 | 1 | 1 | 1 | 1 | 1 | 0.75 | 1 | 1 | 0.75 | 1 | 1 | 1 | 0.75 | 1 | 1 | 1 | 1 | 0.5 |
| 8 | 0.75 | 0.75 | 0.75 | 0.75 | 0.75 | 0.75 | 0.75 | 1 | 0.75 | 0.75 | 1 | 0.75 | 0.75 | 0.75 | 1 | 0.75 | 0.75 | 0.75 | 0.75 | 0.75 |
| 9 | 1 | 1 | 1 | 1 | 1 | 1 | 1 | 0.75 | 1 | 1 | 0.75 | 1 | 1 | 1 | 0.75 | 1 | 1 | 1 | 1 | 0.5 |
| 10 | 1 | 1 | 1 | 1 | 1 | 1 | 1 | 0.75 | 1 | 1 | 0.75 | 1 | 1 | 1 | 0.75 | 1 | 1 | 1 | 1 | 0.5 |
| 11 | 0.75 | 0.75 | 0.75 | 0.75 | 0.75 | 0.75 | 0.75 | 1 | 0.75 | 0.75 | 1 | 0.75 | 0.75 | 0.75 | 1 | 0.75 | 0.75 | 0.75 | 0.75 | 0.75 |
| 12 | 1 | 1 | 1 | 1 | 1 | 1 | 1 | 0.75 | 1 | 1 | 0.75 | 1 | 1 | 1 | 0.75 | 1 | 1 | 1 | 1 | 0.5 |
| 13 | 1 | 1 | 1 | 1 | 1 | 1 | 1 | 0.75 | 1 | 1 | 0.75 | 1 | 1 | 1 | 0.75 | 1 | 1 | 1 | 1 | 0.5 |
| 14 | 1 | 1 | 1 | 1 | 1 | 1 | 1 | 0.75 | 1 | 1 | 0.75 | 1 | 1 | 1 | 0.75 | 1 | 1 | 1 | 1 | 0.5 |
| 15 | 0.75 | 0.75 | 0.75 | 0.75 | 0.75 | 0.75 | 0.75 | 1 | 0.75 | 0.75 | 1 | 0.75 | 0.75 | 0.75 | 1 | 0.75 | 0.75 | 0.75 | 0.75 | 0.75 |
| 16 | 1 | 1 | 1 | 1 | 1 | 1 | 1 | 0.75 | 1 | 1 | 0.75 | 1 | 1 | 1 | 0.75 | 1 | 1 | 1 | 1 | 0.5 |
| 17 | 1 | 1 | 1 | 1 | 1 | 1 | 1 | 0.75 | 1 | 1 | 0.75 | 1 | 1 | 1 | 0.75 | 1 | 1 | 1 | 1 | 0.5 |
| 18 | 1 | 1 | 1 | 1 | 1 | 1 | 1 | 0.75 | 1 | 1 | 0.75 | 1 | 1 | 1 | 0.75 | 1 | 1 | 1 | 1 | 0.5 |
| 19 | 1 | 1 | 1 | 1 | 1 | 1 | 1 | 0.75 | 1 | 1 | 0.75 | 1 | 1 | 1 | 0.75 | 1 | 1 | 1 | 1 | 0.5 |
| 20 | 0.5 | 0.5 | 0.5 | 0.5 | 0.5 | 0.5 | 0.5 | 0.75 | 0.5 | 0.5 | 0.75 | 0.5 | 0.5 | 0.5 | 0.75 | 0.5 | 0.5 | 0.5 | 0.5 | 1 |

As can be seen from Table 5, the 8th, 11th, 15th, and 20th experts have significantly lower weight values than the other experts, which means that when judging the degree of influence on the system $F_1$ itself, the opinions of these experts are clearly inconsistent with other experts, and this situation can also be found from the matrix in Table 3, where the expert 20 with the lowest weight value, for example, he has a consensus degree of 0.75 with only 3 experts and consensus degree with other experts degree are all only 0.5, resulting in his low weight when scoring this element.

Iterating through all factors in turn, the weight matrices $\mathbf{W}^{(n)}, \mathbf{W}_1^{(n)}, \mathbf{W}_2^{(n)}, \mathbf{W}_{1\supset1}^{(n)}, \mathbf{W}_{1\supset2}^{(n)},$ $\mathbf{W}_{1\supset3}^{(n)}, \mathbf{W}_{2\supset1}^{(n)}$ and $\mathbf{W}_{2\supset2}^{(n)}$, where $n = 1,2...,20$, can be obtained for the 20 experts in making decisions on systems $F,F_1,F_2,F_{1\supset1},F_{1\supset2},F_{1\supset3},F_{2\supset1}$ and $F_{2\supset2}$. The weight matrices of expert 1 for each subsystem is given here representatively, as shown in Fig 9.

Due to the limitation of space, detailed data for the remaining expert weighting matrices can be found in **Appendix C** at https://osf.io/gxtj5.

Step 5 Calculate the weighted IDR matrix

According to Eq (20), the weighted IDR matrix of each system is obtained as shown in Fig 10.

Step 6 Solving the super IDR matrix using hierarchical DEMATEL method

Combined with the weighted IDR matrix of each subsystem, the super IDR matrix can be calculated according to the hierarchical DEMATEL method and Eqs (5) to (6) in Section Hierarchical DEMATEL method. The super IDR matrix integrates the mutual influence relationships of all factors, and the calculation results are retained to three decimal places, as shown in Table 6.

Step 7 Bringing the super IDR matrix into the traditional DEMATEL method, using Eqs (1) to (4) can be calculated to obtain the reasonability and centrality of each factor, and non-Archimedean infinitesimal is $\varepsilon = 0.00001$.

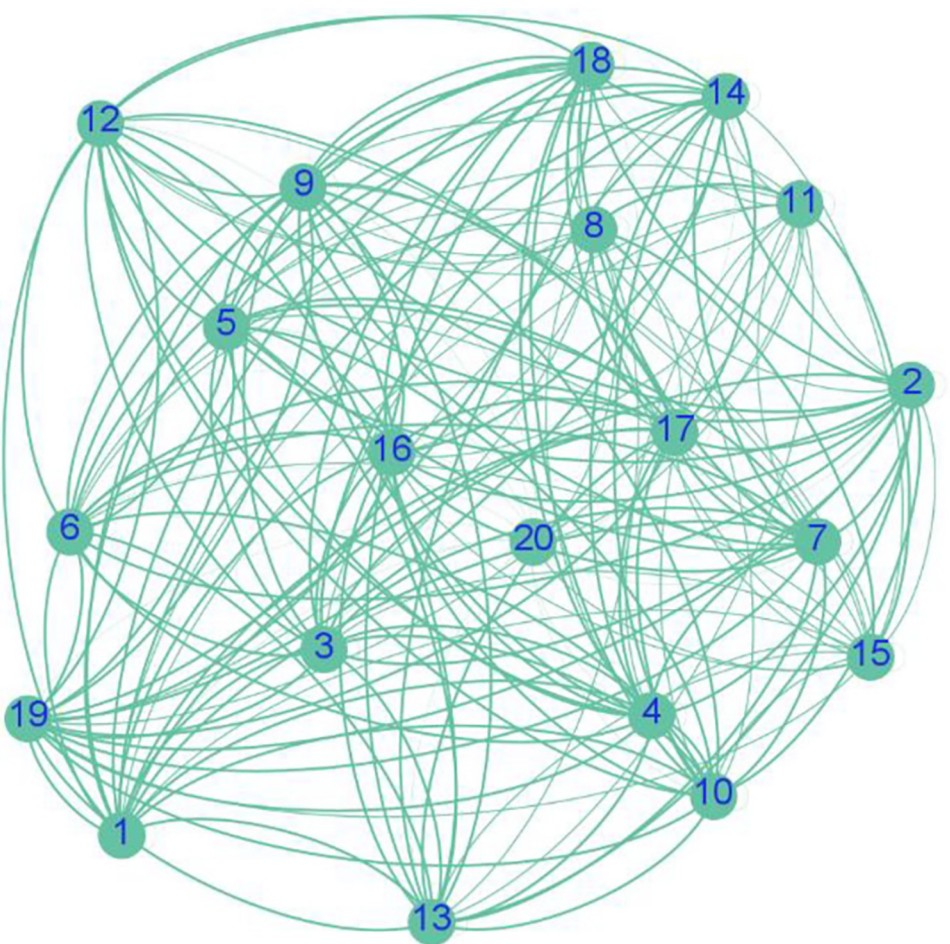

**Fig 8. Expert consistency network.**

As can be seen from Table 7, factors $f_4, f_6, f_9, f_{13}, f_{14}, f_{15}$, and $f_{16}$ are receive factors, and the remaining factors are cause factors, which is consistent with our common sense.

Table 8 shows the centrality of each element, based on which we can further obtain the weights of each element. According to the weights in Table 9, it can be concluded that the importance ranking of factors a influencing combat capability is

$$f_6 \succ f_1 \succ f_7 \succ f_{10} \succ f_{16} \succ f_{11} \succ f_8 \succ f_4 \succ f_5 \succ f_{13} \succ f_{15} \succ f_2 \succ f_{14} \succ f_{12} \succ f_3 \succ f_9$$

It can be seen that the commander's command ability, communication interference and anti-jamming ability, the degree of intelligence of the accusation system, the adequacy of the war reserve, and the speed of attack are the most critical five factors.

**Table 4. Clustering coefficients of experts in network $G_{11}$.**

| Expert | 1 | 2 | 3 | 4 | 5 | 6 | 7 | 8 | 9 | 10 |
|---|---|---|---|---|---|---|---|---|---|---|
| Clustering coefficient | 0.9204 | 0.9204 | 0.9204 | 0.9204 | 0.9204 | 0.9204 | 0.9204 | 0.8806 | 0.9204 | 0.9204 |
| Expert | 11 | 12 | 13 | 14 | 15 | 16 | 17 | 18 | 19 | 20 |
| Clustering coefficient | 0.8806 | 0.9204 | 0.9204 | 0.9204 | 0.8806 | 0.9204 | 0.9204 | 0.9204 | 0.9204 | 0.8488 |

**Table 5. Weights of experts in network $G_{11}$.**

| Expert | 1 | 2 | 3 | 4 | 5 | 6 | 7 | 8 | 9 | 10 |
|---|---|---|---|---|---|---|---|---|---|---|
| Weight | 0.0573 | 0.0573 | 0.0573 | 0.0573 | 0.0573 | 0.0573 | 0.0573 | 0.0237 | 0.0573 | 0.0573 |
| Expert | 11 | 12 | 13 | 14 | 15 | 16 | 17 | 18 | 19 | 20 |
| Weight | 0.0237 | 0.0573 | 0.0573 | 0.0573 | 0.0237 | 0.0573 | 0.0573 | 0.0573 | 0.0573 | 0.0114 |

## Comparative analysis

**Algorithm stability and deviation introduction.** In most studies related to decision-making methods, scholars often analyze the superiority of the proposed method from a subjective perspective by explaining the reasons for differences in decision-making results. However, as decision-making problems themselves are subjective judgment problems, analyzing differences in decision-making results from a subjective perspective cannot provide sufficient evidence for judgment, as the analyst tends to analyze in their favor. This paper provides a new approach by comparing the robustness of decision-making methods in interference environments to analyze the advantages and disadvantages of the methods, i.e., comparing the stability and deviation of decision-making results in the interference of original data. The interference of original data is used to simulate the problem of random deviation in expert scoring caused by interference factors. Although this random deviation is a low-probability event, it can still cause changes in decision-making results when it occurs. Therefore, it is necessary to consider the ability of decision-making algorithms to maintain stability and the degree of deviation towards the true value.

The detailed definitions of the stability and deviation of decision-making algorithms are:

The decision result of the complex system $F = \{f_i | i = 1, 2, \ldots, m\}$ is obtained in descending order of weights $\boldsymbol{\psi} = \begin{bmatrix} f_{a_1} & f_{a_2} & \cdots & f_{a_m} \end{bmatrix}$, and the subscript number of each factor corresponds to the decision vector $\boldsymbol{\eta} = \begin{bmatrix} a_1 & a_2 & \cdots & a_m \end{bmatrix}$. The decision result obtained from the original data in the interference-free condition is denoted as $\boldsymbol{\psi}_0$ and the decision vector is $\boldsymbol{\eta}_0$.

Suppose that $L$ decision experiments are conducted in the interference scenario, and the $L$ decision experiments are independent of each other, and the decision results are $\boldsymbol{\psi}_1, \boldsymbol{\psi}_2, \ldots \boldsymbol{\psi}_L$, and the corresponding decision vectors are $\boldsymbol{\eta}_1, \boldsymbol{\eta}_2, \ldots, \boldsymbol{\eta}_L$.

**Definition 1** Decision algorithm stability

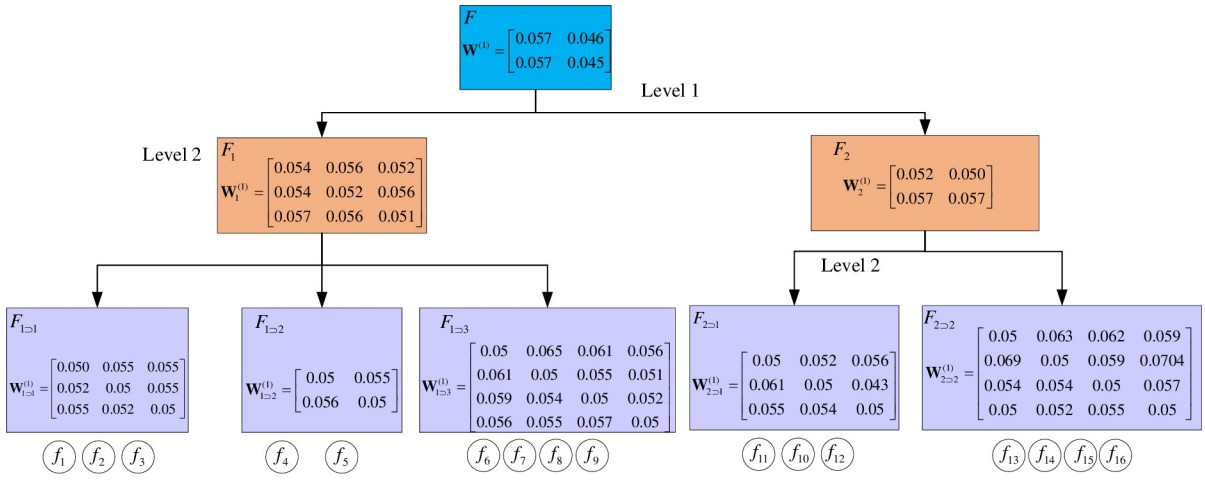

**Fig 9. Weight matrix of expert 1.**

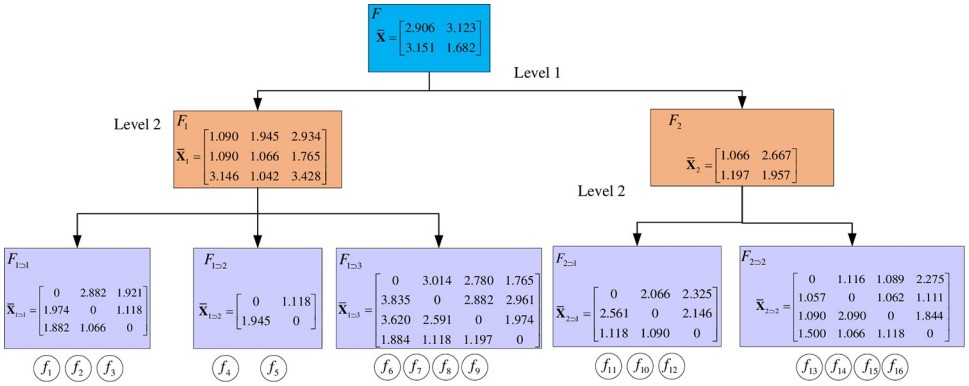

**Fig 10. Weighted IDR matrix.**

The cosine of the angle of the decision vector $\boldsymbol{\eta}_1, \boldsymbol{\eta}_2, \ldots, \boldsymbol{\eta}_L$ in the Euclidean space is considered as the degree of agreement between the two decision outcomes in the $L$ decision experiments, and for the decision vectors $\boldsymbol{\eta}_i$ and $\boldsymbol{\eta}_j$, the degree of agreement between the two vectors is

$$\cos\theta_{ij} = \frac{\boldsymbol{\eta}_i \bullet \boldsymbol{\eta}_j}{|\boldsymbol{\eta}_i||\boldsymbol{\eta}_j|} \tag{21}$$

When $\cos\theta_{ij} = 1$, it means that the decision vector $\boldsymbol{\eta}_i$ is completely consistent with $\boldsymbol{\eta}_j$ and the degree of consistency between the decision vectors has symmetry, i.e., $\cos\theta_{ij} = \cos\theta_{ji}$. From this, we can obtain the consistency matrix of $L$ experiments

$$\mathbf{P} = \begin{bmatrix} \cos\theta_{11} & \cos\theta_{12} & \cdots & \cos\theta_{1L} \\ \cos\theta_{21} & \cdots & \cdots & \cos\theta_{2L} \\ \vdots & \ddots & \cdots & \vdots \\ \cos\theta_{L1} & \cos\theta_{L2} & \cdots & \cos\theta_{LL} \end{bmatrix} \tag{22}$$

**Table 6. Super IDR matrix for combat capability system.**

|  | $f_1$ | $f_2$ | $f_3$ | $f_4$ | $f_5$ | $f_6$ | $f_7$ | $f_8$ | $f_9$ | $f_{10}$ | $f_{11}$ | $f_{12}$ | $f_{13}$ | $f_{14}$ | $f_{15}$ | $f_{16}$ |
|---|---|---|---|---|---|---|---|---|---|---|---|---|---|---|---|---|
| $f_1$ | 0.000 | 0.048 | 0.032 | 0.061 | 0.061 | 0.052 | 0.050 | 0.047 | 0.035 | 0.058 | 0.058 | 0.050 | 0.054 | 0.050 | 0.054 | 0.058 |
| $f_2$ | 0.033 | 0.000 | 0.019 | 0.054 | 0.054 | 0.045 | 0.044 | 0.042 | 0.031 | 0.052 | 0.051 | 0.044 | 0.047 | 0.044 | 0.047 | 0.051 |
| $f_3$ | 0.031 | 0.018 | 0.000 | 0.047 | 0.047 | 0.040 | 0.039 | 0.036 | 0.027 | 0.045 | 0.045 | 0.038 | 0.042 | 0.039 | 0.041 | 0.045 |
| $f_4$ | 0.034 | 0.030 | 0.026 | 0.000 | 0.065 | 0.041 | 0.040 | 0.038 | 0.028 | 0.054 | 0.054 | 0.046 | 0.050 | 0.046 | 0.049 | 0.054 |
| $f_5$ | 0.034 | 0.030 | 0.026 | 0.112 | 0.000 | 0.041 | 0.040 | 0.038 | 0.028 | 0.054 | 0.054 | 0.046 | 0.050 | 0.046 | 0.049 | 0.054 |
| $f_6$ | 0.055 | 0.049 | 0.043 | 0.024 | 0.024 | 0.000 | 0.058 | 0.053 | 0.034 | 0.060 | 0.060 | 0.051 | 0.055 | 0.052 | 0.055 | 0.060 |
| $f_7$ | 0.054 | 0.047 | 0.041 | 0.024 | 0.024 | 0.074 | 0.000 | 0.055 | 0.057 | 0.058 | 0.058 | 0.049 | 0.054 | 0.050 | 0.053 | 0.058 |
| $f_8$ | 0.051 | 0.045 | 0.039 | 0.022 | 0.022 | 0.070 | 0.050 | 0.000 | 0.038 | 0.055 | 0.055 | 0.047 | 0.051 | 0.047 | 0.050 | 0.055 |
| $f_9$ | 0.037 | 0.033 | 0.029 | 0.016 | 0.016 | 0.036 | 0.021 | 0.023 | 0.000 | 0.041 | 0.041 | 0.035 | 0.038 | 0.035 | 0.038 | 0.041 |
| $f_{10}$ | 0.059 | 0.052 | 0.046 | 0.054 | 0.054 | 0.061 | 0.059 | 0.056 | 0.041 | 0.000 | 0.048 | 0.054 | 0.057 | 0.053 | 0.057 | 0.061 |
| $f_{11}$ | 0.059 | 0.052 | 0.045 | 0.054 | 0.054 | 0.061 | 0.059 | 0.055 | 0.041 | 0.059 | 0.000 | 0.049 | 0.057 | 0.052 | 0.057 | 0.060 |
| $f_{12}$ | 0.050 | 0.044 | 0.039 | 0.046 | 0.046 | 0.052 | 0.050 | 0.047 | 0.035 | 0.026 | 0.025 | 0.000 | 0.050 | 0.046 | 0.050 | 0.053 |
| $f_{13}$ | 0.054 | 0.048 | 0.042 | 0.050 | 0.050 | 0.056 | 0.054 | 0.051 | 0.038 | 0.026 | 0.025 | 0.022 | 0.000 | 0.032 | 0.032 | 0.066 |
| $f_{14}$ | 0.051 | 0.045 | 0.039 | 0.047 | 0.047 | 0.052 | 0.051 | 0.048 | 0.036 | 0.024 | 0.023 | 0.021 | 0.031 | 0.000 | 0.031 | 0.032 |
| $f_{15}$ | 0.054 | 0.048 | 0.042 | 0.050 | 0.050 | 0.056 | 0.054 | 0.051 | 0.038 | 0.026 | 0.025 | 0.022 | 0.032 | 0.061 | 0.000 | 0.054 |
| $f_{16}$ | 0.059 | 0.052 | 0.046 | 0.054 | 0.054 | 0.061 | 0.059 | 0.055 | 0.041 | 0.027 | 0.027 | 0.024 | 0.044 | 0.031 | 0.033 | 0.000 |

**Table 7. Reasonability.**

| Factor | $f_1$ | $f_2$ | $f_3$ | $f_4$ | $f_5$ | $f_6$ | $f_7$ | $f_8$ |
|---|---|---|---|---|---|---|---|---|
| Reasonability | 0.383 | 0.128 | 0.201 | -0.391 | 0.267 | -0.429 | 0.184 | 0.001 |
| Factor | $f_9$ | $f_{10}$ | $f_{11}$ | $f_{12}$ | $f_{13}$ | $f_{14}$ | $f_{15}$ | $f_{16}$ |
| Reasonability | -0.500 | 1.055 | 1.199 | 0.426 | -0.457 | -0.800 | -0.273 | -0.994 |

The stability of the algorithm for $L$ experiments is

$$\alpha = \frac{1}{\|\mathbf{P} - \mathbf{I}\|} \tag{23}$$

The matrix $\mathbf{I}$ is a matrix with all elements of 1, $\mathbf{I} \in \mathbf{R}^{L \times L}$, denotes the most ideal case. When $\mathbf{P} = \mathbf{I}$, it means that all the decision results are equal when $L$ experiments are conducted, and the stability of the algorithm is infinite at this time, which means that the decision algorithm is highly stable and the decision results do not change in any random interference scenario.

**Definition 2** Deviation of decision-making algorithms

The decision vector for the $i$ th experiment in $L$ decision experiments is $\mathbf{\eta}_i$, then we have

$$\beta = \frac{\sum_{i=1}^{L} \|\mathbf{\eta}_0 - \mathbf{\eta}_i\|}{L} \tag{24}$$

$\beta$ is the deviation of decision-making algorithms in $L$ decision experiments, which indicates the cumulative deviation of the experimental results from the true value.

As can be seen from the mathematical expressions of stability and deviation, stability mainly describes the degree to which the algorithm keeps the decision results consistent and stable with each other in multiple experiments. When the stability of the algorithm is high, it means that the decision results do not vary much from each other, indicating that the algorithm does not change the decision results easily due to random disturbances. The greater the deviation, the greater the deviation of the decision result from the true value after interference.

For the stability and deviation of a certain algorithm, there are several possibilities:

1. Large stability and large deviation
   Implying that although the results of multiple experiments are stable, they deviate far from the true value.

2. Large stability and small deviation
   Meaning that the results of multiple experiments are stable and each experiment is close to the true value, which is the most desirable situation.

3. Small stability and large deviation
   Meaning that the results of multiple experiments are not stable, and the results deviate from the true value, which is the worst case.

**Table 8. Centrality.**

| Factor | $f_1$ | $f_2$ | $f_3$ | $f_4$ | $f_5$ | $f_6$ | $f_7$ | $f_8$ |
|---|---|---|---|---|---|---|---|---|
| Centrality | 11.299 | 9.964 | 8.734 | 10.431 | 10.424 | 11.640 | 11.274 | 10.646 |
| Factor | $f_9$ | $f_{10}$ | $f_{11}$ | $f_{12}$ | $f_{13}$ | $f_{14}$ | $f_{15}$ | $f_{16}$ |
| Centrality | 7.949 | 11.170 | 11.113 | 9.540 | 10.308 | 9.624 | 10.284 | 11.145 |

**Table 9. Weight.**

| Factor | $f_1$ | $f_2$ | $f_3$ | $f_4$ | $f_5$ | $f_6$ | $f_7$ | $f_8$ |
|--------|-------|-------|-------|-------|-------|-------|-------|-------|
| Weight | 0.0739 | 0.0575 | 0.0441 | 0.0630 | 0.0629 | 0.0784 | 0.0736 | 0.0656 |
| Factor | $f_9$ | $f_{10}$ | $f_{11}$ | $f_{12}$ | $f_{13}$ | $f_{14}$ | $f_{15}$ | $f_{16}$ |
| Weight | 0.0366 | 0.0722 | 0.0715 | 0.0527 | 0.0615 | 0.0536 | 0.0612 | 0.0719 |

4. Small stability and small deviation

Meaning that although the combined results are close to the true value, the algorithm is not stable enough and is susceptible to random interference.

**Comparative analysis of different methods.** To analyze the scientific validity of the method proposed in this paper and its superiority compared to other methods, we compare the fixed-point iteration method for calculating expert weights in reference [36] with the method proposed in this paper. This part continues to use the data in Section Case presentation as the original data, adding data perturbation scenarios for multiple experiments, each experiment is independent of each other, and the data used in both methods in the experiments are guaranteed to be exactly the same after the perturbation.

When the perturbation is zero, the decision results calculated according to the fixed-method of immobile points are

$$f_1 \succ f_6 \succ f_{10} \succ f_{16} \succ f_{11} \succ f_7 \succ f_4 \succ f_5 \succ f_{13} \succ f_8 \succ f_{15} \succ f_{14} \succ f_2 \succ f_{12} \succ f_3 \succ f_9$$

Some differences can be seen with the decision results calculated by the method in this paper, however, we cannot compare the advantages and disadvantages of these two methods by subjective analysis of the decision results only. Therefore, in this section, the analysis is based on the stability and deviation.

We set the perturbation environment as follows:

The extent to which the data are perturbed is controlled by two parameters $p_1$ and $p_2$, $p_1 \in [0,1]$ denotes the proportion of experts that are perturbed and $p_2 \in [0,1]$ denotes the proportion of elements that are perturbed in perturbed experts. As an example, we set $p_1 = 0.3$, $p_2 = 0.2$, $p_1$ means that 30% of the experts will be disturbed, $p_2$ means 20% of these experts' decision data will be randomly perturbed. The perturbed data will be randomly updated to an integer in the range [0–4], and the probability distribution follows a uniform distribution.

In order to consider all cases as much as possible, a range of $p_1 \in 0.04\sim0.36$, $p_2 \in 0.04\sim0.36$ with 0.02 steps was set and a total of 281 sets of experiments were performed considering different combinations of the two parameters. 40 randomized experiments were conducted in each group, and the averaging method, the fixed-point iteration method and the method of this paper were compared and analyzed with the same data and conditions. The averaging method is the control group, and this method treats all experts' weights as equal, and in this paper, each expert's weight is set to 0.05.

From Table 10, it can be seen that inf appears in the second row, indicating that the stability of the algorithm reaches infinity at this point. According to Formula (22), when the results of $L$ independent experiments are exactly the same, matrix $\mathbf{P} = \mathbf{I}$, which means that the stability of the algorithm is infinite, as indicated by Formula (23), and the corresponding offset of the algorithm is 0. This means that throughout the $L$ experimental process, all experiments are exactly the same as the original value.

The first 10 sets of experimental data are given in Table 10, where method 1 refers to the averaging method, method 2 refers to the method in this paper, and method 3 refers to the

Table 10. Experimental data of the first 10 groups.

| | $p_1$ | $p_2$ | Stability | | | Deviation | | |
|---|---|---|---|---|---|---|---|---|
| | | | Method 1 | Method 2 | Method 3 | Method 1 | Method 2 | Method 3 |
| 1 | 0.04 | 0.04 | 0.1962 | 0.5586 | 0.9105 | 12.3023 | 3.8634 | 4.2702 |
| 2 | 0.04 | 0.06 | Inf | Inf | 5.5977 | 0.0000 | 0.0000 | 1.2728 |
| 3 | 0.04 | 0.08 | 0.1685 | 0.5229 | 0.8592 | 18.9306 | 7.1865 | 6.7515 |
| 4 | 0.04 | 0.10 | 0.2167 | 1.1429 | 1.0311 | 13.1443 | 4.2696 | 5.4106 |
| 5 | 0.04 | 0.12 | 1.0304 | 1.6746 | 2.5981 | 3.10650 | 2.7949 | 3.2105 |
| 6 | 0.04 | 0.14 | 0.1630 | 0.8776 | 0.7231 | 17.2712 | 5.6676 | 7.8454 |
| 7 | 0.04 | 0.16 | 0.2496 | 2.4878 | 0.5355 | 9.8888 | 2.4233 | 6.2169 |
| 8 | 0.04 | 0.18 | 0.1848 | 0.3193 | 0.7639 | 16.1442 | 5.3398 | 4.8115 |
| 9 | 0.04 | 0.20 | 0.1624 | 0.6737 | 0.5529 | 21.0647 | 6.1559 | 9.2633 |
| 10 | 0.04 | 0.22 | 0.2220 | 1.0806 | 1.2241 | 13.0083 | 2.1142 | 5.8998 |

Table 11. Data and percentage of optimal performance of the 3 methods.

| | Maximum stability | | Minimal deviation | |
|---|---|---|---|---|
| method | time | percent | time | percent |
| 1 | 1 | 0.35% | 1 | 0.35% |
| 2 | 199 | 68.86% | 236 | 81.66% |
| 3 | 89 | 20.80% | 52 | 17.99% |

method in [36]. Due to the large amount of data, the detailed data of 281 sets of experiments regarding stability and deviation are recorded in **Appendix D** at https://osf.io/gxtj5. According to the results of 281 sets of experiments, the methods with the largest stability and the smallest deviation in each group of experiments were recorded separately, and the number and percentage of occurrences of the three methods were counted, as shown in Table 11.

As can be seen from Table 11, in the comparison of 281 sets of interference experiments, the method proposed in this paper has the largest stability value 199 times, occupying 68.86% of the number of experiments, and the smallest deviation value 236 times, accounting for 81.66% of the total number of experiments. It means that among these three methods, the method of this paper has the highest stability and accuracy, the method in [36] is the second, and the method of averaging directly on the expert decision matrix has the worst stability and accuracy.

## Conclusion

In this study, we proposed a group hierarchical DEMATLE method for the identification of key factors of complex systems. The method inherits the advantages of the hierarchical DEMATLEL method, which can effectively reduce the workload of experts and, at the same time, large-scale group decision making enables more scientific and comprehensive decision results, which involves the expert weight matrix solving method to bring new ideas for the weight calculation when group experts make decisions. The main contributions and innovations of this paper are as follows:

1. Taking into account the expertise and limited knowledge of experts, the experts are assigned weights by factors to measure the overall performance of experts more finely with the weight matrix.

2. The consensus of group experts is described by constructing an expert consistency network, and the degree of consensus of experts is expressed by the assigned clustering coefficients as an important basis for calculating the weights.

3. Stability and deviation indexes are proposed to test the effectiveness of the algorithm, which makes the decision algorithm test more convincing.

In this paper, the proposed method is applied to the identification of key factors of combat capability complex systems, and the proposed method is compared with other methods, and the experimental results have achieved good results. However, this article does not take into account the adjustment of expert opinions, and lacks the decision adjustment and opinion correction process of experts in reaching group consensus. This will be the focus of future research.

## Author Contributions

**Conceptualization:** Weimin Li.

**Data curation:** Tao Zhang.

**Formal analysis:** Tao Zhang.

**Funding acquisition:** Lei Shao.

**Investigation:** Wenyu Chen.

**Methodology:** Wenyu Chen, Lei Shao.

**Writing – review & editing:** Xi Wang.

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
