## [Decision Letter · Decision Letter 0]

10 May 2023

PONE-D-23-09228Large-scale group-hierarchical DEMATEL method for complex systemsPLOS ONE

Dear Dr. Chen,

Thank you for submitting your manuscript to PLOS ONE. After careful consideration, we feel that it has merit but does not fully meet PLOS ONE’s publication criteria as it currently stands. Therefore, we invite you to submit a revised version of the manuscript that addresses the points raised during the review process.

We look forward to receiving your revised manuscript.

Kind regards,

Mehdi Keshavarz-Ghorabaee

Academic Editor

PLOS ONE

Journal Requirements:

   "This research was funded by National Natural Science Foundation of China, grant number 62173339, 61873278."

5. We note you have included a table to which you do not refer in the text of your manuscript. Please ensure that you refer to Table 7 and 9 in your text; if accepted, production will need this reference to link the reader to the Table.

Reviewers' comments:

Reviewer's Responses to Questions

**Comments to the Author**

1. Is the manuscript technically sound, and do the data support the conclusions?

Reviewer #1: Yes

Reviewer #2: Partly

Reviewer #3: Yes

Reviewer #4: Yes

2. Has the statistical analysis been performed appropriately and rigorously? 

Reviewer #1: N/A

Reviewer #2: Yes

Reviewer #3: Yes

Reviewer #4: Yes

3. Have the authors made all data underlying the findings in their manuscript fully available?

Reviewer #1: Yes

Reviewer #2: Yes

Reviewer #3: Yes

Reviewer #4: No

4. Is the manuscript presented in an intelligible fashion and written in standard English?

Reviewer #1: Yes

Reviewer #2: Yes

Reviewer #3: Yes

Reviewer #4: No

5. Review Comments to the Author

Reviewer #1: This paper designs an objective expert weight design method and proposes stability and deviation indexes to test the effectiveness of the method. The idea is interesting and contributions are good. Some revisions need to be conducted as follows:

1. The title, abstract and introduction of this paper all mention that the existing hierarchical DEMATEL method should be extended from small group decision-making situation to large group decision-making situation, but the paper does not seem to reflect the attribute of large group decision-making. Most of the paper introduces the design method of objective expert weight. Although it mentions the use of weighted network clustering coefficient to calculate expert weight, it feels that it is weakly related to large group decision making. Should the topic highlight the innovation of ‘objective weight design’ rather than ‘large-scale’ group decision making? In addition, ‘large-scale’ is a key word in the title, but there is no relevant literature review in the introduction.

2. This paper proposes stability and deviation indexes to test the effectiveness of the method. This method is very good, but does it need to add some advantages of the method proposed in this paper? Because only stability and deviation value cannot reflect the innovation of the method in the aspects of idea design and calculation difficulty.

3. The paper has many details to improve:

(1) When references are quoted, superscript format is set for some references, but not for most of them, so the format is not uniform. For example, only reference ‘ [39] ’in line 372 has the superscript format set.

(2) The first letter of ‘the process of constructing an expert consistency network 344 is shown in Fig. 4.’ in line 344 is not capitalized.

(3) When referring to a section, some sections are capitalized and some are lowercase. For example, line 401 ‘section2.2’ is not capitalized, while most other places are capitalized. It needs to be unified.

(4) When referring to a formula, it has a different expression, for example, line 414 uses ‘equations (9) to (10)’, line 416 uses ‘Eqs. (13) to (16)’, and line 512 uses ‘equations (5)~(6)’. These expressions need to be unified.

(5) The second paragraph ‘Identifying and analyzing the key factors……’ and third paragraph ‘Communication, intelligence, command……’ in Section 4 do not indent the first line.

(6) Some expressions are ambiguous and need further refinement, such as ‘and since it is 20 experts,’ in line 488.

Reviewer #2: 1.The motivation for this study is unclear. From the introduction, it is difficult to find the specific innovation and motivation of this paper. More specifically, the work of this paper is the accumulation of some existing research results. But the in-depth reason for doing this is not clearly stated.

2.In the Large-scale DEMATEL method, there are three articles that you need to consider. You should explain the differences between your article and the existing articles.

A large group linguistic Z-DEMATEL approach for identifying key performance indicators in hospital performance management, Applied Soft Computing.2020.

Large-scale group DEMATEL decision making method from the perspective of complex network, Systems Engineering — Theory & Practice.2021.

A large group hesitant fuzzy linguistic DEMATEL approach for identifying critical success factors in public health emergencies, Aslib Journal of Information Management.2022.

3.The full name references of people in the entire article are all problematic. Only the last name is needed, for example, 'Lufei Huang et al. used DEMATEL to analyze the key elements of circular supply chain management (CSCM) [6]' should be changed to 'Huang et al. used DEMATEL to analyze the key elements of circular supply chain management (CSCM) [6]'.

4.“The traditional DEMATEL method can only be applied to situations with fewer elements. When the number of elements n>10, it will significantly increase the number of expert judgments and workload(lines 124-125).” Based on my review of DEMATEL articles, the number of attributes generally ranges around 11, and some have even up to 20-30.

5.The format of the subheadings is inconsistent (sections 3.4 and 4.1), please check the entire document. Some paragraphs have two spaces between lines, while others do not, please check the entire document. Some formulas are italicized, while others are not (Y_11^1,G_11^1), please standardize the format.

6.The key assumptions on which new model is proposed are missing altogether and its really difficult to assess that how sensitive the results can be to these assumptions.

7.“The large-scale group decision-making is introduced into the hierarchical DEMATEL method, and the applicable expert scale is more than 20 people, which improves the quality of decision-making. (lines 142-144).” Why did the article only select 20 experts? Does this not contradict the concept of large-scale?

8.It is not clear how the experts were selected and the criteria of selection. Data collection should be described.

9.Author(s) should highlight how they determined the model's parameters? the main difficulties can be mentioned.

10.The main findings of the research should be written in conclusion section.

Reviewer #3: This paper presents a hierarchical DEMATEL (Decision-Making Trial and Evaluation Laboratory) method for large-scale group decision-making. I think that the main idea of this paper is interesting. However, I suggest that the authors consider the following comments to improve the paper:

1. I think the paper should be improved by adding a literature review section and citing other MCDM and weighting methods. The author should discuss the popular and recent MCDM methods like CRITIC (CRiteria Importance Through Intercriteria Correlation), Best-Worst Method (BWM), COPRAS (COmplex PRoportional Assessment), WASPAS (Weighted Aggregates Sum Product Assessment), SECA (Simultaneous Evaluation of Criteria and Alternatives), CODAS (COmbinative Distance-based ASsessment), SWARA (Stepwise Weight Assessment Ratio Analysis), MEREC (MEthod based on the Removal Effects of Criteria) and EDAS (Evaluation based on Distance from Average Solution).

2. In the literature review section, the authors should also discuss the recent studies related to the DEMATEL method. Moreover, the main features of the previous studies and the current study should be presented in a table.

3. The structure of the paper should be organized according to the journal requirements.

4. Figures 2 to 4 are not clear. You should improve the presentation of these figures.

5. The framework of the proposed method should be presented in a figure.

6. A discussion section should be added to present the advantages and disadvantages of the proposed method.

7. The manuscript needs to be improved in terms of its use of the English language.

Overall, I think the paper needs to be revised before publication.

Reviewer #4: The manuscript considers the existence of hierarchy with numerous system factors in complex systems, and proposes a hierarchical DEMATEL method for large-scale group decision-making to make DEMATEL better suited for the identification of critical factors in complex systems. Then it is applied to identify and analyze the key factors that influence combat capability, which is a typical complex system, and explain the superiority of the proposed method through comparative analysis. There are certain innovative points in this manuscript. **However, some theoretical errors occurred in some places and there are still several problems that need to be explained or modified. Details are attached.**

6. PLOS authors have the option to publish the peer review history of their article (what does this mean?). If published, this will include your full peer review and any attached files.

Reviewer #1: No

Reviewer #2: No

Reviewer #3: No

Reviewer #4: No

---

## [Author Response · Author response to Decision Letter 0]

22 May 2023

Response to Comments

We would like to express our gratitude to the reviewers and editors for their time and effort in reviewing the manuscript. We have carefully considered all the comments and revised the manuscript accordingly. The major changes can be found in the revised version titled “Revised Manuscript with Track Changes”. In the following section, we provide detailed responses to each comment made by the reviewers. 

Response to Editors

1. Regarding manuscript style

As we could only access the PDF version of the template, we adjusted the format according to the PDF template.

2. Regarding financial status

We have changed the financial statement at the end of the paper to:

This research was funded by National Natural Science Foundation of China, grant number 62173339, 61873278. The funders had no role in study design, data analysis, decision to publish, or preparation of the manuscript.

3. Regarding paper data

The paper data can be found on the website https://osf.io/gxtj5.

4. Regarding the ORCID ID of the corresponding author

We have added the ORCID ID of the corresponding author to the system as requested.

5. Regarding table citation

Thank you for your reminder. We have checked all tables and figures in the revised manuscript, and they are all cited in the text.

Response to Reviewer 1

This paper designs an objective expert weight design method and proposes stability and deviation indexes to test the effectiveness of the method. The idea is interesting and contributions are good. Some revisions need to be conducted as follows:

Comment 1: 

The title, abstract and introduction of this paper all mention that the existing hierarchical DEMATEL method should be extended from small group decision-making situation to large group decision-making situation, but the paper does not seem to reflect the attribute of large group decision-making. Most of the paper introduces the design method of objective expert weight. Although it mentions the use of weighted network clustering coefficient to calculate expert weight, it feels that it is weakly related to large group decision making. Should the topic highlight the innovation of ‘objective weight design’ rather than ‘large-scale’ group decision making? In addition, ‘large-scale’ is a key word in the title, but there is no relevant literature review in the introduction.

Response 1:

Thank you for your suggestion. Your question is very relevant. Regarding your inquiry about the missing attribute of large-scale group decision-making, the original article mentioned that a large-scale group of experts should consist of no less than 20 members. However, this description is too brief. In the revised version, we have added the following characteristics regarding large group decision-making:

Decision making by a large group of experts exhibits the following characteristics:

(1) The group size is relatively large, usually consisting of no fewer than 20 decision-making experts;

(2) The decision-making problem exhibits multidimensional, complex and stochastic attributes;

(3) High consistency requirements need to be met among the group.

When solving problems of large-scale group decision-making, the main difficulties are as follows [35]：

(1) There are significant differences among decision-makers. It is necessary to identify the status of each decision-maker and assign corresponding weights to achieve scientific evaluation results.

(2) Due to the large size of the decision-makers, it is important to use effective methods to gather the opinions of large groups to avoid leverage effects caused by intentional praise or criticism during the evaluation process.

(3) When group opinions are relatively scattered, it is necessary to effectively coordinate the differences in preferences among decision-makers to maximize the satisfaction of large-scale group decisions.

It can be seen that the objective calculation of expert weight is a major difficulty in large group decision making, so the identification of expert weights is investigated in the paper, which is relevant to the study of large group decision making.

Comment 2: 

This paper proposes stability and deviation indexes to test the effectiveness of the method. This method is very good, but does it need to add some advantages of the method proposed in this paper? Because only stability and deviation value cannot reflect the innovation of the method in the aspects of idea design and calculation difficulty.

Response 2:

Thank you for bringing this issue to our attention. We have made some additions to the introduction and conclusion sections of the revised draft to describe the strengths and innovations of the paper. The specific additions are as follows:

The innovations of this article are:

(1) Introducing a new method for identifying the weights of experts in large-scale groups. This method assigns different weights to different indicators, abandoning the practice of using a single weight value to represent the decision-making status of experts under all indicators, in order to address the unique characteristics of each expert in terms of knowledge, skills, experience, and personality.

(2) Using the network clustering coefficient to describe the consistency of expert groups in scoring the same indicator, and calculating expert weights through the consistency between experts and the group, to maximize the requirement of opinion consistency in large-scale group decision-making.

(3) The methods involved are more suitable for analyzing the correlation between various factors within complex systems and identifying key factors. It can not only reduce the workload of experts but also improve the scientificity of decision-making results.

(4) Instead of analyzing the effectiveness of decision-making algorithms through subjective methods as in other studies, this article constructs interference scenarios to analyze the stability and bias of algorithm results when expert decision-making data is interfered with, which is more convincing.

Comment 3: 

The paper has many details to improve:

Comment 3.1: 

(1) When references are quoted, superscript format is set for some references, but not for most of them, so the format is not uniform. For example, only reference ‘ [39] ’in line 372 has the superscript format set.

Response 3.1:

Thank you for your careful suggestion. We have set all references in the text to be in non-superscript form, as required by the journal's template.

Comment 3.2:

(2) The first letter of ‘the process of constructing an expert consistency network 344 is shown in Fig. 4.’ in line 344 is not capitalized.

Response 3.2: 

Thank you for your careful suggestion. We have changed the sentence in the original to “The process of constructing an expert consistency network is shown in Fig. 4”.

Comment 3.3:

(3) When referring to a section, some sections are capitalized and some are lowercase. For example, line 401 ‘section2.2’ is not capitalized, while most other places are capitalized. It needs to be unified.

Response 3.3:

Thank you for your careful suggestion. We have standardised the section you mentioned in the revised version. As the journal's template does not number the section content, we have addressed the section headings directly in the citation and have uniformly capitalised them.

Comment 3.4:

(4) When referring to a formula, it has a different expression, for example, line 414 uses ‘equations (9) to (10)’, line 416 uses ‘Eqs. (13) to (16)’, and line 512 uses ‘equations (5)~(6)’. These expressions need to be unified.

Response 3.4:

Thank you for your suggestion. We have standardised all the expressions in the revised version to the form “equations (a) to (b)”.

Comment 3.5:

(5) The second paragraph ‘Identifying and analyzing the key factors……’ and third paragraph ‘Communication, intelligence, command……’ in Section 4 do not indent the first line.

Response 3.5:

Thank you for your suggestion. We have amended the issues you have mentioned in the revised version.

Comment 3.6:

(6) Some expressions are ambiguous and need further refinement, such as ‘and since it is 20 experts,’ in line 488.

Response 3.6:

Thank you for your careful suggestion. We have changed the sentence in the original to “The weights of the experts are obtained by normalizing the clustering coefficients according to equation (17), the normalization index ，to serve the purpose of reducing the weights of the experts who are far from the group consensus and giving more weights to the experts with high consensus.” 

Response to Reviewer 2

Comment 1: 

The motivation for this study is unclear. From the introduction, it is difficult to find the specific innovation and motivation of this paper. More specifically, the work of this paper is the accumulation of some existing research results. But the in-depth reason for doing this is not clearly stated.

Response 1:

Thank you for pointing out this problem.

In order to more fully explain the motivation for the research in this paper, we have added to the introduction the difficulties in solving the problem of large-scale group decision making and the innovations in this paper. The details are:

When solving problems of large-scale group decision-making, the main difficulties are as follows [35]：

(1) There are significant differences among decision-makers. It is necessary to identify the status of each decision-maker and assign corresponding weights to achieve scientific evaluation results.

(2) Due to the large size of the decision-makers, it is important to use effective methods to gather the opinions of large groups to avoid leverage effects caused by intentional praise or criticism during the evaluation process.

(3) When group opinions are relatively scattered, it is necessary to effectively coordinate the differences in preferences among decision-makers to maximize the satisfaction of large-scale group decisions.

The innovations of this article are:

(1) Introducing a new method for identifying the weights of experts in large-scale groups. This method assigns different weights to different indicators, abandoning the practice of using a single weight value to represent the decision-making status of experts under all indicators. This approach addresses the unique characteristics of each expert in terms of knowledge, skills, experience, and personality.

(2) Using the network clustering coefficient to describe the consistency of expert groups in scoring the same indicator, and calculating expert weights through the consistency between experts and the group. This approach maximizes the requirement of opinion consistency in large-scale group decision-making.

(3) The methods involved are well-suited for analyzing the correlation between various factors within complex systems and identifying key factors. They can not only reduce the workload of experts but also improve the scientificity of decision-making results.

(4) Instead of analyzing the effectiveness of decision-making algorithms through subjective methods, as in other studies, this article constructs interference scenarios to analyze the stability and bias of algorithm results when expert decision-making data is interfered with. This approach is more convincing.

Comment 2:

In the Large-scale DEMATEL method, there are three articles that you need to consider. You should explain the differences between your article and the existing articles.

[1] A large group linguistic Z-DEMATEL approach for identifying key performance indicators in hospital performance management, Applied Soft Computing.2020.

[2] Large-scale group DEMATEL decision making method from the perspective of complex network, Systems Engineering — Theory & Practice.2021.

[3] A large group hesitant fuzzy linguistic DEMATEL approach for identifying critical success factors in public health emergencies, Aslib Journal of Information Management.2022.

Response 2:

Thank you for your careful reminder and your question is very professional.

The research ideas and methods in literature (3) and (1) are very similar, with the difference being mainly in the linguistic scale functions used, and the main research ideas used are largely the same.

(1) In references [1] and [3], the weights of expert clusters were calculated using the maximum consensus method. The process of solving the weight optimization problem by maximizing consensus is essentially an optimization process, and the relative size relationship of the weights of each group as an optimization constraint is entirely based on the subjective experience of experts. Both papers have come to the conclusion that "The sensitivity analysis shows that the weights of clusters can have a big influence on the final ranking of performance indicators." This suggests that the results of such decisions still depend on biases towards a certain group, rather than presenting results as objectively as possible for the entire large-scale group.

In our paper, we did not determine the weights of expert clusters, but instead measured the weight of each individual based on the difference in clustering coefficients relative to all experts. The expert weight is determined entirely based on the expert's scoring performance, without adding any subjective bias information.

(2) Obviously, the three papers mentioned above did not use the hierarchical DEMATEL method. The hierarchical DEMATEL method we used can reduce the workload of experts. Taking the case in reference [3] as an example, the factors involved mainly include:

There are 15 factors. By using the methods of the above-mentioned three papers, experts need to make 210 judgments when constructing the direct influencing matrix. In addition, experts need to make a choice from seven levels (s0-s6) when making each judgment. To put it mildly, it would be difficult for a conscientious expert to make 1470 rigorous decisions assuming they are careful enough.

In contrast, the hierarchical DEMATEL method used in our paper requires experts to make only judgments when constructing the direct influencing matrix for the same case. According to our paper's 0-4 scaling method, experts only need to make 345 judgments. This does not impose a decision-making burden on experts, and the advantage becomes more apparent as the number of factors in the system increases.

(3) The above three references did not take into account the decision-making limitations caused by differences in experts' professional backgrounds and knowledge. Although reference [2] also mentioned the calculation method of expert weights, it still adopted the method of using a single weight value to represent the expert's status in all fields, which is the same as most researches without determining the expert's weight according to different domains. The proposed method in this article constructs the weight values of experts in different professions. A certain expert may have a higher weight value when making decisions on a certain indicator, but a lower weight value when making decisions on another indicator. Each expert has different weight values under different factors, which can enhance the weight of experts in some small fields and make the weight values more targeted, thus increasing the scientificity of decision-making.

(4) Compared to the literature mentioned above, this paper adds analysis indicators of algorithm deviation and stability, and proposes the superiority of the proposed method through constructing data interference scenarios. Although sensitivity analysis was performed in literature [1] [3], it mainly analyzed the influence of clustering weights on decision results. It can be seen that decision results are only affected by clustering weights, which are constrained by expert qualitative judgments. Therefore, there is a lack of analysis methods for the stability of the decision algorithm itself.

Comment 3:

The full name references of people in the entire article are all problematic. Only the last name is needed, for example, 'Lufei Huang et al. used DEMATEL to analyze the key elements of circular supply chain management (CSCM) [6]' should be changed to 'Huang et al. used DEMATEL to analyze the key elements of circular supply chain management (CSCM) [6]'.

Response 3:

Thank you very much for your suggestion. We realize the problems in our writing and have made modifications according to your advice.

Comment 4: 

“The traditional DEMATEL method can only be applied to situations with fewer elements. When the number of elements n>10, it will significantly increase the number of expert judgments and workload (lines 124-125).” Based on my review of DEMATEL articles, the number of attributes generally ranges around 11, and some have even up to 20-30.

Response 4:

Thank you for your inquiry. This article introduces the hierarchical DEMATEL method mainly to reduce the workload of experts, especially in the context of complex systems or complex organizational decision-making problems involving multiple factors with clear hierarchical characteristics. 

Complex systems have three distinctive features, numerous system factors, various types of impacts, and the existence of hierarchical structures. For a system with N factors, the original decomposition method requires experts to make judgments N×(N-1) times. In addition, assuming H types of influences between numerous factors in a complex system, it takes H×N×(N-1) time to make judgments. Clearly, as the number of system factors and influence types increases, the judgment time will exponentially increase, making it difficult to make accurate judgments.

As mentioned in our response to your second question, using the traditional DEMATEL method to calculate the case in reference [3] would result in a significant workload for experts despite there only being 15 factors involved in the case. However, using the hierarchical DEMATEL method can effectively reduce the burden on expert decision-making, and this advantage becomes more significant as the number of factors in the system increases. 

Therefore, introducing the hierarchical DEMATEL method is meant to reduce the workload on experts and avoid potential psychological fatigue and feelings of exhaustion due to excessive workload. While existing methods can theoretically address cases with many decision-making factors, the issue of high workload for experts must not be ignored when considering practical expert decision-making situations. 

I hope my answer will be satisfactory.

Comment 5:

The format of the subheadings is inconsistent (sections 3.4 and 4.1), please check the entire document. Some paragraphs have two spaces between lines, while others do not, please check the entire document. Some formulas are italicized, while others are not (Y_11^1,G_11^1), please standardize the format.

Response 5:

Thank you for your suggestion. Regarding the formatting issue, we have made uniform adjustments in the revised draft. Also, we have discovered issues such as the absence of indents for the first line in the second paragraph "Identifying and analyzing the key factors..." and the third paragraph "Communication, intelligence, command..." in Section 4.

We have unified the formula format issue in the revised manuscript. The unified rule is as follows: When a quantity is a matrix or vector, it should be presented in upright and bold; when a quantity is a scalar or variable, it should be presented in italic and non-bold.

Comment 6:

The key assumptions on which new model is proposed are missing altogether and its really difficult to assess that how sensitive the results can be to these assumptions.

Response 6:

Thank you for your inquiry.

[1] A large group linguistic Z-DEMATEL approach for identifying key performance indicators in hospital performance management, Applied Soft Computing.2020.

[3] A large group hesitant fuzzy linguistic DEMATEL approach for identifying critical success factors in public health emergencies, Aslib Journal of Information Management.2022.

Regarding key assumptions and sensitivity analysis, both references [1] and [3] involve sensitivity analysis of key assumptions on decision outcomes, mainly because in their studies, assumptions were made about the weights of expert initial clustering, which were based on expert qualitative understanding.

However, in this paper, as the weights of experts are solely based on the consistency judgment between expert decision results, no assumptions about any weights are necessary, thus there are no key assumptions, and all weights are calculated based on the consistency network of expert decision results.

Nevertheless, we also considered the sensitivity of the decision algorithm, i.e., the extent to which the decision results are affected when the input variables change. In our paper, this is presented in the form of the algorithm's "stability", and we compared it with two other methods, concluding that our proposed method has the highest stability.

Thank you for your suggestion, and I hope my answer satisfies you.

Comment 7:

“The large-scale group decision-making is introduced into the hierarchical DEMATEL method, and the applicable expert scale is more than 20 people, which improves the quality of decision-making. (lines 142-144).” Why did the article only select 20 experts? Does this not contradict the concept of large-scale?

Response 7:

Thank you for your questioning. We realize there was an error in the wording of the paper. With regards to the number of experts in large-scale groups, many literatures have given a consistent definition, such as the literature you mentioned "Large-scale group DEMATEL decision making method from the perspective of complex network, Systems Engineering — Theory & Practice.2021" which indicates that "generally, when the number of experts is greater than or equal to 20, it is called a large-scale group." The statement in our paper "more than 20 people" is not rigorous and should be amended to "not less than 20 people". We have made corresponding modifications to our paper.

Comment 8:

It is not clear how the experts were selected and the criteria of selection. Data collection should be described.

Response 8:

Thank you for your suggestion. We have provided explanations regarding the experts' backgrounds and criteria, as well as data collection methods in Section Case presentation and Methodology analysis. The 20 experts mainly consist of military theory researchers, weapon equipment professionals, and combat commanders.

Comment 9:

Author(s) should highlight how they determined the model's parameters? the main difficulties can be mentioned.

Response 9:

In this paper, the network clustering coefficient is a key parameter involved. However, regarding the calculation of the directed weighted network clustering coefficient, relevant research has already been quite mature, so this paper cites the research results of others.

As for the determination of other model parameters, this paper does not involve them. All parameters are determined based on existing methods. The innovation of this paper mainly lies in method innovation and extension, so the determination of parameters is not the main content. However, I appreciate your suggestion, and it will be a key issue for the author's subsequent attention.

I hope my answer will be satisfactory.

Comment 10: 

The main findings of the research should be written in conclusion section.

Response 10:

Thank you for your suggestion. We have supplemented the findings, advantages, disadvantages and next research directions in the conclusion section. The revised conclusion is as follows:

In this study, we proposed a group hierarchical DEMATLE method for the identification of key factors of complex systems. The method inherits the advantages of the hierarchical DEMATLEL method, which can effectively reduce the workload of experts and, at the same time, large-scale group decision making enables more scientific and comprehensive decision results, which involves the expert weight matrix solving method to bring new ideas for the weight calculation when group experts make decisions. The main contributions and innovations of this paper are as follows:

(1) Taking into account the expertise and limited knowledge of experts, the experts are assigned weights by factors to measure the overall performance of experts more finely with the weight matrix.

(2) The consensus of group experts is described by constructing an expert consistency network, and the degree of consensus of experts is expressed by the assigned clustering coefficients as an important basis for calculating the weights.

(3) Stability and deviation indexes are proposed to test the effectiveness of the algorithm, which makes the decision algorithm test more convincing.

In this paper, the proposed method is applied to the identification of key factors of combat capability complex systems, and the proposed method is compared with other methods, and the experimental results have achieved good results. However, this article does not take into account the adjustment of expert opinions, and lacks the decision adjustment and opinion correction process of experts in reaching group consensus. This will be the focus of future research.

Response to Reviewer 3

This paper presents a hierarchical DEMATEL (Decision-Making Trial and Evaluation Laboratory) method for large-scale group decision-making. I think that the main idea of this paper is interesting. However, I suggest that the authors consider the following comments to improve the paper:

Comment 1:

I think the paper should be improved by adding a literature review section and citing other MCDM and weighting methods. The author should discuss the popular and recent MCDM methods like CRITIC (Criteria Importance Through Intercriteria Correlation), Best-Worst Method (BWM), COPRAS (Complex PRoportional Assessment), WASPAS (Weighted Aggregates Sum Product Assessment), SECA (Simultaneous Evaluation of Criteria and Alternatives), CODAS (Combinative Distance-based Assessment), SWARA (Stepwise Weight Assessment Ratio Analysis), MEREC (Method based on the Removal Effects of Criteria) and EDAS (Evaluation based on Distance from Average Solution).

Response 1:

Thank you for your suggestions. We have added a discussion of these methods in the introduction section of the article and inserted some new references. The additional references are [4]-[13].

Comment 2:

In the literature review section, the authors should also discuss the recent studies related to the DEMATEL method. Moreover, the main features of the previous studies and the current study should be presented in a table.

Response 2:

Thank you for your suggestion. We have created a table in the introduction section that describes the latest research involving the DEMATEL method, comparing its application scenarios, innovative means, and methods.

Table 1 Application of the DEMATEL method

Papers Method Application Scenario Innovative approach

[15] [21]

Classic DEMATEL Investigate the role of human factors in promoting the establishment of sustainable continuous improvement (SCI) environment; Identify the key factors affecting the supply chain in the electronics industry Application of classical method

[16] [20]

AHP+DEMATEL Assessing critical success factors for circular supply chain management (CSCM) implementation of blockchain; Explore the key factors influencing stock price behavior Methods composition application

[17]

IFS+DEMATEL Analyses have been conducted on the critical challenges of the COVID-19 vaccine supply chain Methods composition application

[18]

BWM+BN+DEMATEL Identifying the impact of risk factors and sources of information on the decision-making process Methods composition application

[19] [28]

Gray DEMATEL Studying the causal relationships of influencing factors in the decision-making process Methods composition application

[22] [25]

Fuzzy DEMATEL Estimate and map the suitability classes of ecotourism potentials in the study area of "Dunayski kljuc" region (Serbia)；Analyzing the facilitating factors for supply chain responsiveness Methods composition application

[26]

Gray DEMATEL+ANP Explores favorable methods to evaluate the green mining performance (GMP) of underground gold mines Methods composition application

[30]

DEMATEL A new matrix normalization method has been researched and proposed Innovation in Method

[31] [32]

Hierarchical DEMATEL The hierarchical DEMATEL method has been proposed to make the DEMATEL method applicable to complex systems with many factors; based on the proposed hierarchical DEMATEL method, a program for small-group experts to reach consensus has been designed Innovation in Method

Comment 3: 

The structure of the paper should be organized according to the journal requirements.

Response 3:

Thank you for your suggestion. We have downloaded the PDF template of the journal according to the requirements and adjusted the structure of the article in accordance with the template.

Comment 4: 

Figures 2 to 4 are not clear. You should improve the presentation of these figures. 

Response 4:

Thank you for your suggestion. We have redrawn figures 2, 3 and 4, and enlarged the unclear numbers in the original images.

Comment 5: 

The framework of the proposed method should be presented in a figure.

Response 5:

Thank you for your suggestion. We believe that your suggestion is excellent, as it allows for a more intuitive depiction of the main steps in our decision algorithm. The added flowchart is shown below:

Fig. 5. Flow chart of the decision algorithm

Comment 6: 

A discussion section should be added to present the advantages and disadvantages of the proposed method.

Response 6:

Thank you for your suggestion. We have supplemented the findings, advantages, disadvantages and next research directions in the conclusion section. The revised conclusion is as follows:

In this study, we proposed a group hierarchical DEMATLE method for the identification of key factors of complex systems. The method inherits the advantages of the hierarchical DEMATLEL method, which can effectively reduce the workload of experts and, at the same time, large-scale group decision making enables more scientific and comprehensive decision results, which involves the expert weight matrix solving method to bring new ideas for the weight calculation when group experts make decisions. The main contributions and innovations of this paper are as follows:

(1) Taking into account the expertise and limited knowledge of experts, the experts are assigned weights by factors to measure the overall performance of experts more finely with the weight matrix.

(2) The consensus of group experts is described by constructing an expert consistency network, and the degree of consensus of experts is expressed by the assigned clustering coefficients as an important basis for calculating the weights.

(3) Stability and deviation indexes are proposed to test the effectiveness of the algorithm, which makes the decision algorithm test more convincing.

In this paper, the proposed method is applied to the identification of key factors of combat capability complex systems, and the proposed method is compared with other methods, and the experimental results have achieved good results. However, this article does not take into account the adjustment of expert opinions, and lacks the decision adjustment and opinion correction process of experts in reaching group consensus. This will be the focus of future research.

Comment 7: 

The manuscript needs to be improved in terms of its use of the English language.

Response 7:

Thank you for your suggestion. We have checked the English grammar throughout the entire text and invited colleagues who are native speakers of American English to proofread it.

Response to Reviewer 4

The manuscript considers the existence of hierarchy with numerous system factors in complex systems, and proposes a hierarchical DEMATEL method for large-scale group decision-making to make DEMATEL better suited for the identification of critical factors in complex systems. Then it is applied to identify and analyze the key factors that influence combat capability, which is a typical complex system, and explain the superiority of the proposed method through comparative analysis. There are certain innovative points in this manuscript. However, some theoretical errors occurred in some places and there are still several problems that need to be explained or modified.

Comment 1: 

The definition of horizontal decomposition and vertical decomposition in Line 229 need to be explained briefly, and the case used in this manuscript does not involve the horizontal decomposition part, so it is recommended to reconsider whether to keep this part. 

Response 1:

Thank you for your suggestion. We have added a discussion on vertical and horizontal decomposition in the Section Hierarchical DEMATEL Method of our revised manuscript：

Vertical decomposition refers to splitting a complex function into multiple single and simple functions based on functional attributes. Horizontal decomposition refers to hierarchically dividing a system according to its hierarchical characteristics.

Regarding the doubt you mentioned about whether the case study involves horizontal decomposition, in the case study, the process of classifying combat capabilities is not only based on functional attributes, but also includes hierarchical classification. Therefore, both forms of classification are involved in the case study.

I hope my answer will be satisfactory.

Comment 2: 

The combat capability system shown in Fig 5 is a three-level structure according to the theory of Yuanwei Du’s paper. Why is the manuscript explained as a two-level structure in Line 452? 

Response 2:

Thank you for your suggestion.

If the combat capability is taken into consideration, it is a three-level structure; if not, it is a two-level structure.

As shown in the figure below, in Yuanwei Du's paper case, the highest-level was not counted as a layer, so it is the same as our article, the combat capability system should be a two-layer structure.

Comment 3:

Also in Fig 5, 、 should be modified to 、 since the system only has two subsystems in level 1.

Response 3:

Thank you for pointing out our mistake. We did not notice this error when writing. We appreciate it very much. We have made changes to Figure 5 (which is now Figure 6 in the revised version). 

Comment 4: 

As can be seen from Table 4, the weight of expert 1 is 0.0573 when judging the degree of the influence on the system itself. Why does the weight change to 0.054 in Fig 8?

Response 4:

According to your reminder, I have checked the data in the paper again and found no errors. It was my mistake that I wrote the wrong network name during writing. The network used in the paper should be formed when scoring the first element in the first row and column of system , not .

Therefore, the weight corresponding to Expert 1 should be the first element of matrix in Figure 8 (which is now Figure 9 in the revised version), which is 0.057 after rounding to three decimal places.

The cause of the above problem is that I mistakenly used the data of network to represent network . Thank you for patiently and carefully discovering this problem. I have corrected the incorrect description of and related parts in the paper. The corrected data are consistent with each other and consistent with the data in the supplementary materials submitted earlier.

Comment 5: 

There may be a misunderstanding of “based on which 20% of these experts' decision data will be randomly perturbed” in L605-606. What does “these experts” refer to? “20% of these experts' decision data will be randomly perturbed” means that the data of 30% of the experts who have been disturbed will be perturbed or means that the data of all the experts involved in scoring will be perturbed?

Response 5:

Thank you for your question, it shows that we are not expressing ourselves clearly enough in our writing.

Our intention is：The data of 30% of the experts who have been disturbed will be perturbed.

We have rephrased the statement in the revised draft：

we set , means that 30% of the experts will be disturbed, means 20% of these experts' decision data will be randomly perturbed.

Comment 6: 

In the second row of Table 9, when , the indicators of Method 1 and Method 2 are significantly different from other situations, and it is recommended to provide a reasonable explanation.

Response 6:

Thank you for your suggestion. We have added the following content to the article for clarification: 

From Table 10, it can be seen that inf appears in the second row, indicating that the stability of the algorithm reaches infinity at this point. According to formula (22), when the results of independent experiments are exactly the same, matrix , which means that the stability of the algorithm is infinite, as indicated by Formula ([Disp-formula pone.0288326.e112]), and the corresponding offset of the algorithm is 0. This means that throughout the experimental process, all experiments are exactly the same as the original value.

Comment 7: 

The conclusion section should add future research work.

Response 7:

Thank you for your suggestion. We have supplemented the findings, advantages, disadvantages and next research directions in the conclusion section. The revised conclusion is as follows:

In this study, we proposed a group hierarchical DEMATLE method for the identification of key factors of complex systems. The method inherits the advantages of the hierarchical DEMATLEL method, which can effectively reduce the workload of experts and, at the same time, large-scale group decision making enables more scientific and comprehensive decision results, which involves the expert weight matrix solving method to bring new ideas for the weight calculation when group experts make decisions. The main contributions and innovations of this paper are as follows:

(1) Taking into account the expertise and limited knowledge of experts, the experts are assigned weights by factors to measure the overall performance of experts more finely with the weight matrix.

(2) The consensus of group experts is described by constructing an expert consistency network, and the degree of consensus of experts is expressed by the assigned clustering coefficients as an important basis for calculating the weights.

(3) Stability and deviation indexes are proposed to test the effectiveness of the algorithm, which makes the decision algorithm test more convincing.

In this paper, the proposed method is applied to the identification of key factors of combat capability complex systems, and the proposed method is compared with other methods, and the experimental results have achieved good results. However, this article does not take into account the adjustment of expert opinions, and lacks the decision adjustment and opinion correction process of experts in reaching group consensus. This will be the focus of future research.

Comment 8: 

English language of this manuscript should be improved. Please carefully proofread your manuscript to improve its presentation and readability. For instance, the phrase “For example” appears multiple times, with two consecutive lines appearing in Line 287 and Line 288.

Response 8: 

Thank you for your suggestion. We have checked the English grammar throughout the entire text and invited colleagues who are native speakers of American English to proofread it.

We have processed the issue you mentioned about the excessive use of "for example" and deleted the second redundant one.

Comment 9:

Besides, some format errors should be corrected in this manuscript.

1.The first line of each paragraph should be indented, while Line 433、Line 440 and Line 487 do not.

Response 9.1:

Thank you for your careful suggestion. We have made revisions to the formatting issues in the revised draft.

2.Tables 6、7、8 are not three-line tables.

Response 9.2:

Thank you for your suggestion. In the revised draft, we have corrected Tables 6, 7, and 8 to three-line tables. As an additional table has been added, the modified tables are now Tables 7, 8, and 9.

3.Definition 1 in Line 552 and Definition 2 in Line 567 are not bolded while others are bolded. Please keep consistence.

Response 9.3:

Thank you for your reminder. We have highlighted these contents in bold when revising the draft.

4.The subheading 4.2.2 indents while others do not.

Response 9.4:

Thank you for your suggestions. We have reorganized all the titles in the article, removed section numbers as required by the template, and standardized the title format. We also changed the original 4.2.2 title "Comparative Analysis" to "Comparative Analysis of Different Methods" to avoid confusion.

5.The right border of the 10th group in three-line table of Table 9 needs to be removed.

Response 9.5:

Thank you for the reminder. We have made the necessary adjustments based on your feedback.

6.The references contain an excessive number of formatting errors. For example, references [21]、[22]and [36].

Response 9.6:

Thank you for your suggestion. We have made modifications to the inappropriate citation format in the introduction section of the manuscript.

---

## [Decision Letter · Decision Letter 1]

18 Jun 2023

PONE-D-23-09228R1Large-scale group-hierarchical DEMATEL method for complex systemsPLOS ONE

Dear Dr. Chen,

Thank you for submitting your manuscript to PLOS ONE. After careful consideration, we feel that it has merit but does not fully meet PLOS ONE’s publication criteria as it currently stands. Therefore, we invite you to submit a revised version of the manuscript that addresses the points raised during the review process.

We look forward to receiving your revised manuscript.

Kind regards,

Mehdi Keshavarz-Ghorabaee

Academic Editor

PLOS ONE

Journal Requirements:

Reviewers' comments:

Reviewer's Responses to Questions

**Comments to the Author**

1. If the authors have adequately addressed your comments raised in a previous round of review and you feel that this manuscript is now acceptable for publication, you may indicate that here to bypass the “Comments to the Author” section, enter your conflict of interest statement in the “Confidential to Editor” section, and submit your "Accept" recommendation.

Reviewer #1: All comments have been addressed

Reviewer #2: All comments have been addressed

Reviewer #4: (No Response)

2. Is the manuscript technically sound, and do the data support the conclusions?

Reviewer #1: Yes

Reviewer #2: Partly

Reviewer #4: Yes

3. Has the statistical analysis been performed appropriately and rigorously? 

Reviewer #1: N/A

Reviewer #2: N/A

Reviewer #4: N/A

4. Have the authors made all data underlying the findings in their manuscript fully available?

Reviewer #1: Yes

Reviewer #2: Yes

Reviewer #4: Yes

5. Is the manuscript presented in an intelligible fashion and written in standard English?

Reviewer #1: Yes

Reviewer #2: Yes

Reviewer #4: Yes

6. Review Comments to the Author

Reviewer #1: The paper has been revised according to my suggestions. I suggest that the editor consider accepting this paper.

Reviewer #2: The manuscript is much more readable after revision. Furthermore, the authors implemented the reviewer comments, carefully. Thus, the paper can be accepted regarding my suggestion.

**Reviewer #4: The revised manuscript improved a lot both in writing and formatting. However, there are still several problems that need to be explained or modified. The details see attachment.**

7. PLOS authors have the option to publish the peer review history of their article (what does this mean?). If published, this will include your full peer review and any attached files.

Reviewer #1: No

Reviewer #2: No

Reviewer #4: No

---

## [Author Response · Author response to Decision Letter 1]

19 Jun 2023

Response to Comments

We would like to express our gratitude to the reviewers and editors for their time and effort in reviewing the manuscript. We have carefully considered all the comments and revised the manuscript accordingly. The major changes can be found in the revised version titled “Revised Manuscript with Track Changes”. In the following section, we provide detailed responses to each comment made by the reviewers.

Response to Reviewer 4

The revised manuscript improved a lot both in writing and formatting. However, there are still several problems that need to be explained or modified.

Response: 

Thank you for your recognition of our manuscript revision work. With regard to the issues you mentioned this time, we reply to each of them as follows.

Comment 1: 

Table 1 is not three-line table.

Response 1:

Thank you for your reminder. We have revised Table 1 and the sections that span two pages have been dealt with by way of a continuation table.

Comment 2: 

English language should be improved. For instance, the phrase “In summary” appears in two consecutive paragraphs in Line 99 and Line 103.

Response 2:

Thank you for your suggestion. There was a problem with semantic repetition in the original formulation in Line 99 and Line 103, and we have redacted the repetition. Thank you for identifying this problem.

Comment 3: 

The understanding of vertical decomposition and horizontal decomposition in this manuscript is different from Du’s paper which you refer to. The manuscript argues that “Vertical decomposition refers to splitting a complex function into multiple single and simple functions based on functional attributes. Horizontal decomposition refers to hierarchically dividing a system according to its hierarchical characteristics”. However, Du points that horizontal decomposition focuses on dividing the critical factor identification problem of complex systems into several simple problems and vertical decomposition focuses on dividing the complex system into multi-level subsystems under a specific rule. Horizontal decomposition provides the rules for making vertical decomposition.

Response 3:

Thank you for your careful analysis. In order to avoid misunderstandings to the reader and to respect the original meaning of the references made, we have modified the discussion of vertical and horizontal decomposition in the text according to the meaning given in the references.

Comment 4:

In line 337, the manuscript uses in Fig.2 to explain the subscript, but Fig.2 does not include .

Response 4:

Thank you very much for pointing this out. Our intention was to use to illustrate Figure 2, but it was an oversight on our part that led to this error.

In the revised version, we have changed to to illustrate Figure 2.

Comment 5:

There is a doubt about the combat capability system shown in Fig.6 is a three-level structure. The first level are and , the second level are the 5 subsystems of communication, intelligence, command, logistics and fire support, and the third level are the factors , which is a component of F and cannot be decomposed.

Response 5:

Thank you for your statement and comments. Our views are in agreement with yours. Figure 6 represents the secondary structure.

You commented on this issue during the first revision and we responded with the following:

As shown in the figure below, in Yuanwei Du's paper case, the highest-level was not counted as a layer, so it is the same as our article, the combat capability system should be a two-layer structure.

We have therefore clarified this in the revised draft at the time of the first rework, and in our discussion of Figure 6, the discussion reads (Line 504 to Line 508 in the original).

The combat capability system is decomposed into a two-level structure according to the hierarchy, with the first level containing two subsystems, command and control and communications , and firepower and logistical support . The second level contains 5 subsystems of communication , intelligence , command , logistics , and fire support . These five subsystems specifically contain these 16 specific factors.

Therefore, the text is consistent with your view. However, Figure 6 does look like a Three-level structure and has been redrawn for a clearer representation, with the parts of the presentation hierarchy highlighted to avoid misleading the reader. The revised Figure 6 is shown below:

Fig. 6. Hierarchy of combat capability

---

## [Decision Letter · Decision Letter 2]

26 Jun 2023

Large-scale group-hierarchical DEMATEL method for complex systems

PONE-D-23-09228R2

Dear Dr. Chen,

We’re pleased to inform you that your manuscript has been judged scientifically suitable for publication and will be formally accepted for publication once it meets all outstanding technical requirements.

Kind regards,

Mehdi Keshavarz-Ghorabaee

Academic Editor

PLOS ONE

Additional Editor Comments (optional):

Reviewers' comments:

Reviewer's Responses to Questions

**Comments to the Author**

1. If the authors have adequately addressed your comments raised in a previous round of review and you feel that this manuscript is now acceptable for publication, you may indicate that here to bypass the “Comments to the Author” section, enter your conflict of interest statement in the “Confidential to Editor” section, and submit your "Accept" recommendation.

Reviewer #4: All comments have been addressed

2. Is the manuscript technically sound, and do the data support the conclusions?

Reviewer #4: Yes

3. Has the statistical analysis been performed appropriately and rigorously? 

Reviewer #4: Yes

4. Have the authors made all data underlying the findings in their manuscript fully available?

Reviewer #4: Yes

5. Is the manuscript presented in an intelligible fashion and written in standard English?

Reviewer #4: Yes

6. Review Comments to the Author

Reviewer #4: I am satisfied with the revised version. My comments including the description of Table 1, the improvements of English language of the manuscript and the understanding of vertical decomposition and horizontal decomposition have been addressed clearly. Some mistakes have been corrected.

7. PLOS authors have the option to publish the peer review history of their article (what does this mean?). If published, this will include your full peer review and any attached files.

Reviewer #4: No

---

## [Editor Report · Acceptance letter]

7 Jul 2023

PONE-D-23-09228R2 

Large-scale group-hierarchical DEMATEL method for complex systems 

Dear Dr. Chen:

I'm pleased to inform you that your manuscript has been deemed suitable for publication in PLOS ONE. Congratulations! Your manuscript is now with our production department. 

Kind regards, 

on behalf of

Dr. Mehdi Keshavarz-Ghorabaee 

Academic Editor

PLOS ONE